

# Current knowledge of the Southern Hemisphere marine microbiome in eukaryotic hosts and the Strait of Magellan surface microbiome project

Manuel Ochoa-Sánchez[1,2,3], Eliana Paola Acuña Gomez[1], Lia Ramírez-Fenández[4,5], Luis E. Eguiarte[2] and Valeria Souza[1,2]

[1] Centro de Estudios del Cuaternario de Fuego, Patagonia y Antártica (CEQUA), Punta Arenas, Chile
[2] Instituto de Ecología, Universidad Nacional Autónoma de México, Ciudad de México, México
[3] Posgrado en Ciencias Biológicas, Universidad Nacional Autónoma de México, Ciudad de México, México
[4] Facultad de Recursos Naturales Renovables, Universidad Arturo Prat, Iquique, Chile
[5] Centro de Desarrollo de Biotecnología Industrial y Bioproductos, Antofagasta, Chile

Corresponding author
Valeria Souza,
souza.valeria2@gmail.com

## ABSTRACT

Host-microbe interactions are ubiquitous and play important roles in host biology, ecology, and evolution. Yet, host-microbe research has focused on inland species, whereas marine hosts and their associated microbes remain largely unexplored, especially in developing countries in the Southern Hemisphere. Here, we review the current knowledge of marine host microbiomes in the Southern Hemisphere. Our results revealed important biases in marine host species sampling for studies conducted in the Southern Hemisphere, where sponges and marine mammals have received the greatest attention. Sponge-associated microbes vary greatly across geographic regions and species. Nevertheless, besides taxonomic heterogeneity, sponge microbiomes have functional consistency, whereas geography and aging are important drivers of marine mammal microbiomes. Seabird and macroalgal microbiomes in the Southern Hemisphere were also common. Most seabird microbiome has focused on feces, whereas macroalgal microbiome has focused on the epibiotic community. Important drivers of seabird fecal microbiome are aging, sex, and species-specific factors. In contrast, host-derived deterministic factors drive the macroalgal epibiotic microbiome, in a process known as "microbial gardening". In turn, marine invertebrates (especially crustaceans) and fish microbiomes have received less attention in the Southern Hemisphere. In general, the predominant approach to study host marine microbiomes has been the sequencing of the 16S rRNA gene. Interestingly, there are some marine holobiont studies (*i.e.*, studies that simultaneously analyze host (*e.g.*, genomics, transcriptomics) and microbiome (*e.g.*, 16S rRNA gene, metagenome) traits), but only in some marine invertebrates and macroalgae from Africa and Australia. Finally, we introduce an ongoing project on the surface microbiome of key species in the Strait of Magellan. This is an international project that will provide novel microbiome information of several species in the Strait of Magellan. In the short-term, the project will improve our knowledge about microbial diversity in the region, while long-term potential benefits

include the use of these data to assess host-microbial responses to the Anthropocene derived climate change.

# INTRODUCTION

The Southern Hemisphere, particularly the Southern Ocean and its associated ecosystems, is characterized by its unique biodiversity (*Rogers et al., 2020*; *Gutt et al., 2021*). These important ecosystems are facing major abiotic challenges as climate change intensifies. These challenges are primarily driven by ocean warming and increased UV radiation (*Thompson & Solomon, 2002*; *Swart et al., 2018*). Sea surface warming creates stronger water column stratification (*Pellichero et al., 2017*), as well as higher variability in the duration and extent of sea ice sheet and increased glacier melt rate in the southernmost regions (*Gutt et al., 2015*; *Comiso et al., 2017*). Additionally, ocean warming increases microplankton metabolic activity, which in turn accelerates oxygen depletion in the water column (*Schmidtko, Stramma & Visbeck, 2017*), and decreases ocean pH (*McNeil & Matear, 2008*). Tragically, recent intense wildfires and volcanic eruptions in the Southern Hemisphere increased the Antarctic ozone hole size in 2020—2021, which is expected to worsen ocean warming effects (*Yook, Thompson & Solomon, 2022*). Conversely, atypical glacier melt rate is stimulating marine primary productivity, creating complex scenarios in ice dependent species (*Piñones & Fedorov, 2016*). For example, Antarctic krill (*Euphausia superba*), a key species in the Antarctic trophic network, requires ice in its early stages, while also foraging in areas with high chlorophyll *a* concentration (*Kawaguchi et al., 2006*). Current evidence suggests that habitat quality heterogeneity along its Antarctic distribution will produce contractions in krill distribution (*Atkinson et al., 2019*; *Veytia et al., 2020*, but see *Cox et al., 2018*).

However, climate change effects could also spur marine productivity, which might have cascading effects on the trophic network. For example, warmer temperatures coupled with low to moderate winds increase ice melt, which in turn increases iron release (*Hodson et al., 2017*). Iron is a primary productivity limiting factor, so its increased availability triggers diatom growth, which in turn increases krill recruitment (*Noble et al., 2013*; *Bertrand et al., 2007*, *2015*). Ultimately, an increase in krill biomass provides greater resources for predators, which overall increases energy transfer along the trophic network (*Saba et al., 2014*). As climate change progresses, an intensification in seasonality is expected, which might intensify alterations in biological processes (*e.g.*, bottom-up mechanisms).

For instance, in the Strait of Magellan – the southernmost continental region of South America – climate change-derived effects have been recorded since the second half of the XX and early XXI century. These include increasing sea surface temperature (*Smith & Reynolds, 2004*) and higher glacier melt rate (*Aniya, 1999*; *Dixon & Ambinakudige, 2015*).

In the terrestrial ecosystem, warmer seasonality is expected to increase aridity in the Patagonian region, particularly in areas with herbaceous vegetation (*Soto-Rogel et al., 2020*). Nevertheless, there have been no formal studies in the region regarding the effect of climate change on any of its ecosystem properties (*e.g.*, trophic network interactions, biogeochemical cycles, and environmental status). This is unfortunate since the region offers an invaluable geographic position. From the marine perspective, the Strait of Magellan is uniquely influenced by the Pacific and Atlantic oceans, as well as the Cape Horn Current. Additionally, glacier melt seasonal input creates local primary productivity bursts that have bottom-up effects that recruit species from higher trophic status, which increases the biodiversity of the region.

Despite all the previous natural history studies and museums related to macro-organismal diversity, eukaryotes live inside a wider microbial world. Eukaryotic homeostasis (*e.g.*, physiology, immunology, and metabolism) is driven or at least greatly influenced by microbes (*McFall-Ngai et al., 2013*; *Cani et al., 2019*; *Peixoto, Harkins & Nelson, 2021*). As a result, the "hologenome evolution theory" and "holobiont theory" emerged (*Zilber-Rosenberg & Rosenberg, 2008*; *Bordenstein & Theis, 2015*). Holobiont is not a new term *per se*. It was first introduced by Lynn Margulis to describe the biological unit formed between a host and a single inherited endosymbiont (*Margulis, 1991*). The novelty of holobiont arose because of the development and reduced cost of next-generation sequencing (NGS) technologies, which has spurred host microbial community research. Studies have revealed that microbes are ubiquitous in every single metazoan (*Simon et al., 2019*). Thus, the current concept of holobiont refers to a cohesive evolutionary unit formed by the host and its associated microbes (*Bordenstein & Theis, 2015*; *Rosenberg & Zilber-Rosenberg, 2018*). In other words, a holobiont is a single ecological unit comprising an intricate network of mutualistic, commensalism, and parasitic relationships between microbes and their host that are critical for the survival of all organisms involved.

However, microbial influence in the holobiont might vary among hosts (*Hammer, Sanders & Fierer, 2019*). Moreover, there are several examples where hosts do not have a consistent microbial composition across individuals: mainly insects, such as ants (*Sanders et al., 2017*), caterpillars (*Hammer et al., 2017*), solitary bees (*Kwong et al., 2017*), herbivorous beetles (*Kelley & Dobler, 2011*), but also in vertebrates, such as birds (*Hird et al., 2015*) and pandas (*Xue et al., 2015*). These examples were interpreted as flaws in the holobiont concept, encouraging its dismissal since the microbiome is a transient reflect of the environment without any evolutionary relevance to its host (*Moran & Sloan, 2015*; *Douglas & Werren, 2016*). However, recent evidence encourages the reevaluation of the role that microbiomes and holobionts play in ecology and evolution of species (*Koide, 2023*). The holobiont concept has been recently regarded as a useful concept where specific and consistent microbiome compositions underlie host fitness, such as in microcrustaceans *Daphnia* (*Callens et al., 2018*), corals (*Brener-Raffalli et al., 2018*), and plants (*Vannier et al., 2018*), and may have evolutionary relevance for these hosts. Here, we do not discuss inherent evolutive properties of holobionts, instead we argue that holobiont is a useful term to describe the genetic information from the host and its associated

microbes, which could come from random environmental or ecological-evolutive processes (*Theis et al., 2016*). In any case, the first issue regarding holobiont research is the characterization of host microbiomes (*Simon et al., 2019*).

In marine hosts—which are unlikely to be studied using experimental designs—holobiont research has focus on the characterization of the host associated microbes in a relevant environmental and evolutionary framework (*Leray et al., 2021*). To date, metabarcoding has been the most popular method to tackle this issue although it only serves as an initial step to characterize the microbial composition. To test the holobiont hypothesis (*i.e.*, host and associated microbes as a unit), additional steps encompassing host associated microbe functional characterization (*e.g.*, metagenomic or metatranscriptomic data), coupled with host molecular information (*e.g.*, genomic or transcriptomic data), and relevant environmental variables measurement (*e.g.*, seasonal replicates and environmental metadata) must be performed to test the congruence of the holobiont response against seasonal fluctuations.

Once we have baseline information about bacterial composition associated with marine hosts, we should be able to recognize hosts whose microbial communities are far from the expected natural variability. Changes in the natural composition of host microbial communities are collectively termed *dysbiosis* (*Zaneveld, McMinds & Vega Thurber, 2017*). Thus, the microbiome itself could be used as a biosensor of host status (*Zolti et al., 2020*; *Inda & Lu, 2020*). In marine species, there are several examples of the interplay between host fitness and microbial symbiosis, especially in sponges (*Pita et al., 2018* and references therein) and algae (*Van der Loos, Eriksson & Falcão Salles, 2019* and references therein). Dysbiosis is the microbial fingerprint underlying the disruption of the host health and ecology, a highly relevant topic in the Anthropocene.

Microbial communities routinely colonize metazoan internal (*e.g.*, gut, oral) or external (*e.g.*, skin) tissues (*Ross, Rodrigues Hoffmann & Neufeld, 2019*; *Diaz & Reese, 2021*; *ANID R20F0009, 2020*). However, they differ in the selective pressures that influence their assembly; while internal microbial community assembly is influenced by diet and host physiology, external microbial community assembly is greatly influenced by environmental perturbations that impair host fitness (*Byrd, Belkaid & Segre, 2018*; *Kuziel & Rakoff-Nahoum, 2022*). Therefore, we think that eukaryotic epibiotic microbial communities could serve as valuable tools to survey environmental status.

The epidermis/outer surface of eukaryotes is considered a hostile environment, yet it is frequently colonized by microbes. These microbes must cope with constant shedding (in epidermis)/molt (in feathers), intense solar radiation exposure, low temperature, pH changes, and antimicrobial molecules (*Percival et al., 2012*). Nevertheless, skin microbes play important roles in the host health (*Apprill et al., 2014*), since they are the first line defense against pathogens and actively participate in the host immune system maturation (*Ross, Rodrigues Hoffmann & Neufeld, 2019*).

Marine eukaryotes vary greatly in the nature and complexity of their superficial tissue; therefore, the nature of the surface/skin is critical when studying the marine skin microbiome of any host. For example, algae and fish both have a mucus layer on their surface/skin, but their composition and function are different (*Gomez, Sunyer & Salinas,*

*2013*; *Van der Loos, Eriksson & Falcão Salles, 2019*). In birds and mammals, the epidermis is covered with feathers and hair, respectively. Importantly, seabirds and marine mammals have developed different strategies to cope with freezing water; therefore, their epidermis and associated elements (*i.e.*, feathers or hair) are completely different from species inhabiting tropical and temperate latitudes (*Ross, Rodrigues Hoffmann & Neufeld, 2019*). Conversely, crustaceans solely have a chitin exoskeleton outer surface, so their adaptations to cold temperatures should be studied in their microbiome and physiology. Even with the unique biodiversity living in the Southern Hemisphere, few holobiont studies have been conducted in the region. Biodiversity studies are becoming increasingly important as climate change imposes major threats on ecosystems worldwide, especially in cold environments.

To date, marine holobiont studies have had spatial and phylogenetic biases: on one hand, most studies have been conducted in the tropics or in the Northern hemisphere (most of the references referring to marine hosts in *Ross, Rodrigues Hoffmann & Neufeld (2019)*, *Garrido-Cardenas & Manzano-Agugliaro (2017)*); on the other hand, the most studied species have been sessile organisms, like sponges. Therefore, marine host microbiome research in the Southern Hemisphere has received less attention, particularly in South America. Here, we review the marine host-microbiota/microbiome interactions occurring in the Southern Hemisphere. This review will serve as a diagnosis of the field progress in the region and to detect knowledge gaps and opportunities for further research. Thus, it is intended for scientists interested in eukaryote-associated microbes and using microbiota/microbiomes as biosensors of eukaryotic health.

In this review, our main objectives are to (i) review the current knowledge regarding marine holobiont and microbiota/microbiome studies in the Southern Hemisphere, and (ii) describe an ongoing long-term project that will improve knowledge of microbial communities associated with selected taxa in the Strait of Magellan, Chile (ANID R20F00009). We start with a brief overview of the surface characteristics of marine hosts considered in this review. We then review microbiome patterns and holobiont interactions following trophic level of each marine host. We begin with primary producers (macroalgae (*Macrocystis pyrifera*)), then with primary and secondary animal consumers (channel sprawn (*Munida gregaria*), centolla (*Lithodes santolla*), humpback whale (*Megaptera novaeangliae*), Magellanic (*Spheniscus magellanicus*) and king penguins (*Aptenodytes patagonicus*)), finishing with top predators (South American sea lion (*Otaria byronia*)). Finally, we describe the ongoing project "Surface microbiome of key species in the Strait of Magellan" and bring our perspectives in the field.

## Survey methodology

We covered all marine microbe-eukaryote interaction reports on the Southern Hemisphere that we were able to find. For this, we performed a comprehensive literature analysis of studies spanning the last two decades in the following online databases: PubMed, Science Direct, Scopus and Google Scholar. The search was concluded on February 1, 2023. Only studies in English were selected for further inspection. The following keywords were used to perform the literature search in combination with

the terms, -holobiont-, -microbiota- and -microbiome-: seaweed, sponge, invertebrate, crustacean, marine vertebrate, whales, seals, seabirds. Articles matching any of these words were examined to affirm that sampling was performed in hosts inhabiting the Southern Hemisphere. Although an extensive set of keywords were used, only 80 articles were included in our review. We acknowledge that our findings might have some limitations specifically for invertebrates, such as corals and jellyfish.

## RESULTS

Overall, geographical distribution of marine host-microbial research has important biases. There are geographical biases in sampling efforts across the Southern Hemisphere. For instance, marine microbiome research has been done predominantly in the Antarctic (above 60° S), especially in the Western Antarctic, and temperate latitudes (between 20—40° S), especially in Australia. In contrast, tropical (between 0—20° S) and cold (between 40—60° S) latitudes, as well as South America and Africa have received less attention (Fig. 1). There are also host biases, both in terms of number of studies and species studied. In terms of number of studies per group, marine mammals are the most studied group with 22 studies, followed by sponges (16), seabirds (15), macroalgae (13), invertebrates (10), and fishes (four). Interestingly, marine mammals and sponges, the two most studied groups, have been studied across all latitudinal regions and continents in the Southern Hemisphere (Fig. 2). However, when it came to the number of species studied, the pattern changed somewhat. While sponges remain as the most studied group with 65 studied species, the other important groups were macroalgae (24 studied species) and marine invertebrates (23 studied species) (Table 1). In general, the predominant molecular approach to study host-associated microbial communities are taxonomic markers (*i.e.*, 16S rRNA gene, hereafter 16S) (Fig. 3). Shotgun metagenomics, a microbiome functional approach, has also been used, especially in Australia and Antarctic hosts, whereas South American marine hosts have not been studied with any functional approach to date (Fig. 3, Table 1). However, we acknowledge that our keyword selection could hindered our ability to detect shotgun metagenomic approaches in marine hosts in South America. In any case, if any study of this type has been performed there are scarce.

### Macroalgae as an ecosystem

Marine macroalgae are important ecosystem engineers that play critical roles in primary productivity, biogeochemical cycles, and biodiversity recruitment in marine ecosystems (*Tuya, Wernberg & Thomsen, 2008*). There are 11,017 macroalgal species with cosmopolitan distribution (*Guiry & Guiry, 2023*). However, up to date only the microbiome of 24 macroalgal species have been studied in the Southern Hemisphere (Table 1). Therefore, macroalgal microbiome from most species remains unknown. Most studies were conducted using 16S but there were microbiome approaches using shotgun metagenomics and those that simultaneously address host genetic and microbiome DNA traits (*i.e.*, DNA holobiont (Table 1)). Macroalgal microbiome sampling has been conducted in all continents from the Southern Hemisphere, predominantly in Australia, Antarctica, and South America (Fig. 2).
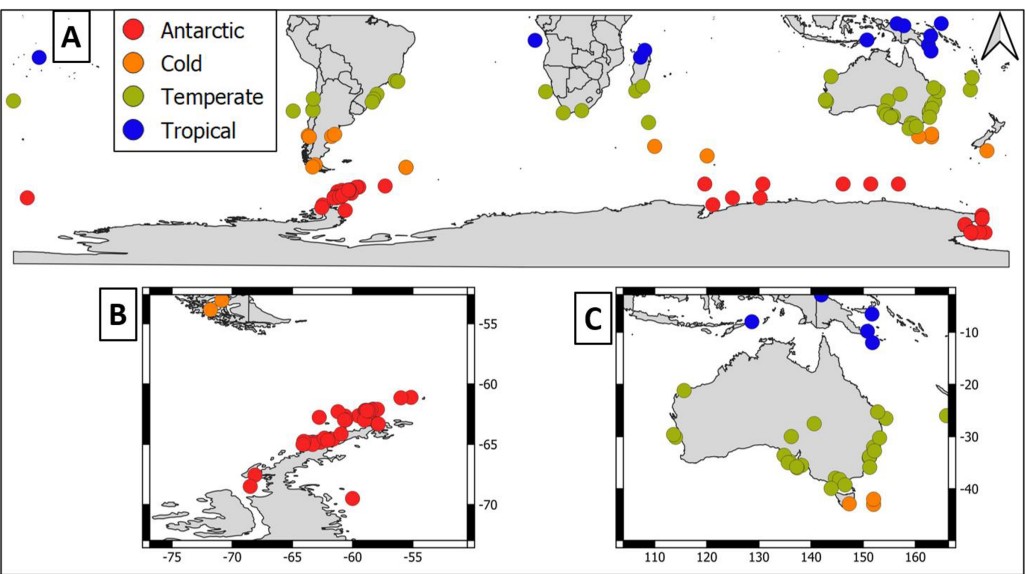

**Figure 1 Latitudinal distribution of marine microbiome studies in the Southern Hemisphere.** (A) Worldwide Southern Hemisphere. Insets with highly sampled regions. (B) Western Antarctic Peninsula. (C) Australia.

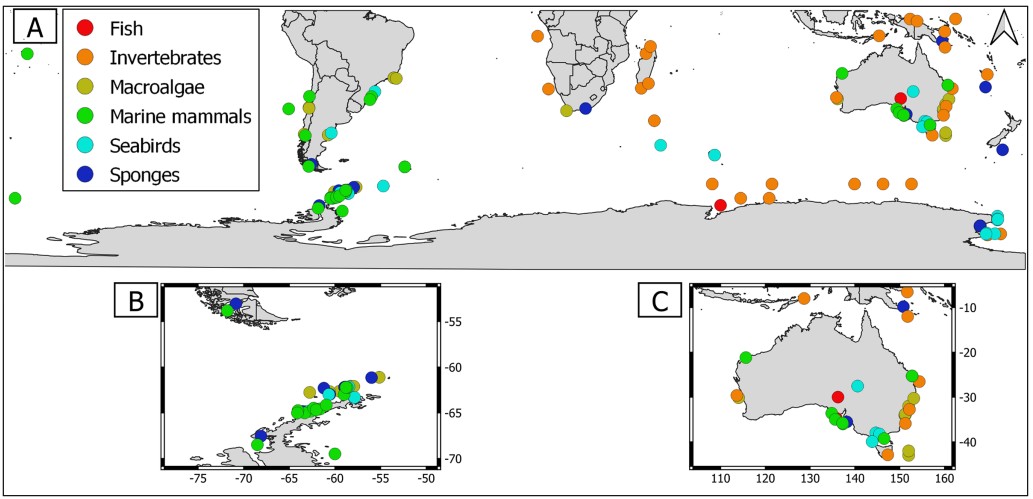

**Figure 2 Distribution of marine host microbiome studies in the Southern Hemisphere.** (A) Worldwide Southern Hemisphere. Insets with highly sampled regions. (B) Western Antarctic Peninsula. (C) Australia.

Algae surfaces are idoneous niches for aerobic and polymer degrading bacteria since their photosynthetic activity and rich composition in carbon and nutrients (such as agar, carrageenan, and cellulose) facilitate the establishment of aerobic and heterotrophic bacteria (*de Oliveira et al., 2012*). Microbial macroalgal epibiotic communities (biofilms) have been extensively studied in the tropics and the Northern Hemisphere (*Malik et al., 2020* and references therein). Additionally, several macroalgae biofilm studies have been conducted in the Southern Hemisphere, revealing that Cyanobacteria, Proteobacteria, Firmicutes, Bacteroidetes and Actinobacteria were the prevalent bacterial phyla

**Table 1 Summary of marine host microbiome research sampling effort in the Southern Hemisphere.**

| Number of species sampled | Species repetitively studied (number of studies) | Predominant molecular approach | Functional microbiome/holobiont approaches | Regions repetitively sampled (number of studies) |
|---|---|---|---|---|
| **Macroalgae** | | | | |
| 24 | *Ecklonia radiata* (4) | 16S | DNA holobiont (*Wood et al., 2022*) Shotgun metagenomics (*Song et al., 2021*) | Antarctica (3) Australia (5) |
| **Sponges** | | | | |
| 65 | *Mycale acerate* (4) | 16s | Shotgun metagenomics (*Moreno-Pino et al., 2020, 2021, Yang et al., 2022*) | Antarctica (13) |
| **Marine invertebrates** | | | | |
| 23 | *Euphasia superba* (2) | 16S | Shotgun metagenomics (*Oh et al., 2021*) DNA & RNA holobiont (*Lan et al., 2021*; *Osvatic et al., 2023*) | Antarctica (4) Australia (3) |
| **Fish** | | | | |
| 9 | NA | 16S | NA | Australia (2) |
| **Seabirds** | | | | |
| 22 | *Aptenodytes patagonicus* (2) *Eudyptula minor* (2) *Pygoscelis adeliae* (3) *Pygoscelis antarcticus* (4) *Pygoscelis papua* (4) *Spheniscus magellanicus* (2) | 16S | Metatranscriptomics (*Marcelino et al., 2019*) | Antarctica (5) Australia (4) Bird Island, South Georgia (2) |
| **Marine mammals** | | | | |
| 15 | *Megaptera novaeanglia* (6) *Balaenoptera musculus* (2) *Neophoca cinerea* (2) *Mirounga leonine* (2) | 16S | Shotgun metagenomics (*Lavery et al., 2012*) | Antarctica (4) Australia (6) Chile (2) |

(*de Oliveira et al., 2012*; *Albakosh et al., 2016*; *Gaitan-Espitia & Schmid, 2020*). Fungi are also important players in macroalgal biofilm composition in the Southern Hemisphere, and different species have been documented from Antarctic seaweeds. The most prevalent fungi associated to macroalgal biofilm were the filamentous fungus *Pseudogymnoascus pannorum*, and the yeast *Metschnikowia australis* (*Loque et al., 2010*; *Godinho et al., 2013*; *Furbino et al., 2017*; *Ogaki et al., 2019*).

## Global knowledge of macroalgae microbiome

There has been extensive research on the macroalgae holobiont that provides a rich body of evidence on the study of macroalgae biofilms. These studies have shown that bacteria and fungi inhabiting the macroalgae biofilm actively interact with their host to influence growth, development, and immune function (*Van der Loos, Eriksson & Falcão Salles, 2019*). Valuable microbes to the algal host appear to be taxonomically restricted to bacteria at higher taxonomic levels. Therefore, biofilm composition may be redundant at the

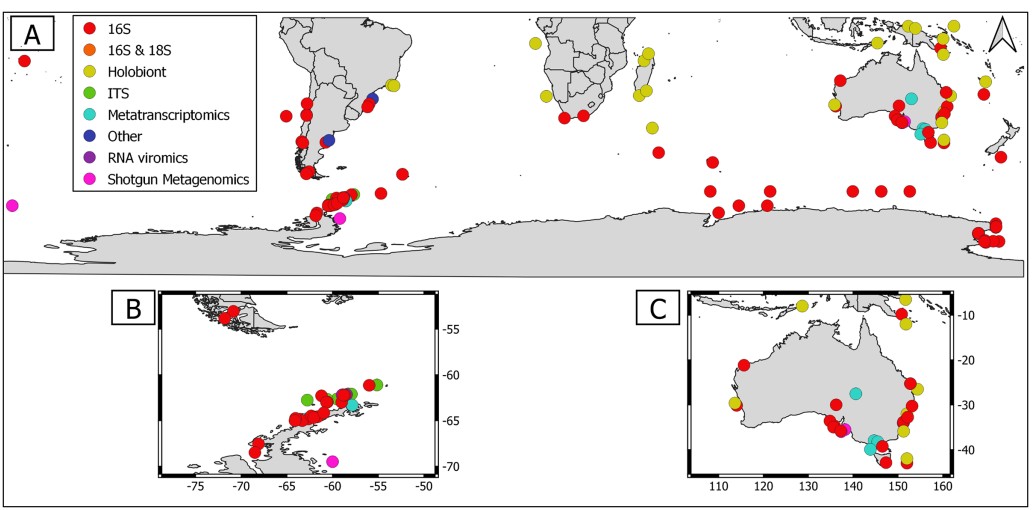

**Figure 3 Distribution of molecular approaches used to study marine host microbiome studies in the Southern hemisphere.** (A) Worldwide Southern Hemisphere. Insets with highly sampled regions. (B) Western Antarctic Peninsula. (C) Australia.

phylum or class level. Nevertheless, at lower taxonomic levels (genus/Amplicon Sequence Variant (ASV)/Operational Taxonomic Unit (OTU)), microbes that positively interact with macroalgae are variable (*Egan et al., 2013*; *Hollants et al., 2013*). Despite taxonomic variability in bacterial genera inhabiting biofilms, traits among bacteria are shared, creating biofilms with different taxonomic composition but with similar functions (*Egan et al., 2013*).

Macroalgae seem to actively recruit their biofilm composition, in a process known as "microbial gardening" (*Saha & Weinberger, 2019*). Microbial gardening is the recruitment of specific beneficial bacteria to the algae. The recruited microbes release antibiotics, quorum sensing inhibitors, and digestive vesicles (*Wiese et al., 2009*; *Romero et al., 2010*; *Richards et al., 2017*) that collectively shape biofilm composition. Although algal biofilm composition varies taxonomically in space, time, and host (*Lachnit et al., 2011*), it shares important traits related to algal morphogenesis (*Wiese et al., 2009*) and nutrient supplementation (*Hollants et al., 2013*). Overall, "correct" microbial gardening might produce a particular biofilm composition capable of producing a cocktail of metabolites with defensive properties against bacterial pathogens, fouling by diatoms (*Saha & Weinberger, 2019*), or predators, such as barnacle larvae or mussels (*Saha, Goecke & Bhadury, 2018*). Interestingly, the benefit conferred to algae by the surface microbiome is not taxonomically restricted, suggesting that microbial traits underlying algal defense are shared by several bacterial taxa.

Some bacteria in the algal biofilm produce antibiotic compounds, which act as a microbial filter for the establishment of environmental microbes (*Albakosh et al., 2016*). For example, algal thallus produces dimethylsulfoniopropionate (*Kessler et al., 2018*), which recruits the bacterial genus *Roseovarius*. Recruited bacteria releases specific morphogenetic compounds that enable correct algal morphogenesis. Moreover, bacterial metabolites could enhance algal performance during stress. Thallusin, a microbial-derived

metabolite, positively influences algal growth, cell differentiation, cell wall development, and rhizoid formation during abiotic stress (*Alsufyani et al., 2020*). Therefore, thallusin production might be an excellent example of a holobiont environmental stress response.

Globally, algal holobiont will be subject to complex scenarios under climate change. Increased sea surface temperature and $CO_2$ concentration might have different outcomes in algal species and their biofilm composition, which in turn, could impair host health (*Gaitan-Espitia & Schmid, 2020*; *Marzinelli et al., 2015*). Moreover, it is imperative to consider the effect on the interactions among microbes and microbe-algae. For example, algal hosts sensitive to acidification might experience lower photosynthetic rates, which might hamper aerobic bacterial proliferation (*Van der Loos, Eriksson & Falcão Salles, 2019*). Climate change-driven environmental perturbations might produce common dysbiotic biofilm composition in stressed macroalgae (*Marzinelli et al., 2015*). Whether shifts in macroalgal biofilm composition associated to environmental perturbation impairs macroalgal fitness remains unexplored.

## Macroalgal holobiont in the Southern Hemisphere

To the best of our knowledge macroalgal microbiome has been addressed in 13 studies in the Southern Hemisphere. Most of these studies have been conducted in temperate latitudes, particularly in Australia (Fig. 2). There are important gaps in tropical and cold latitudes, as well as in selected regions, like South America, Africa, and Indonesia (Fig. 2). Moreover, we still ignore South America native macroalgal microbiome since there was only one study that addressed this topic in a native species, *Macrocystis pyrifera* (Laminariales: Laminariaceae), while another explored the biofilm composition of a macroalgal invasive species, *Undaria pinnitafida* (Laminariales: Alariaceae) (*Florez et al., 2019*; *Lozada et al., 2022*). From these examples, we know that macroalgal biofilm composition is different from seawater and is influenced by seasonality and available nutrients. Interestingly, in the invasive macroalgae *U. pinnitafida* in the Southern Hemisphere, its biofilm composition is dominated by one gammaproteobacterium of the genus *Leucothrix* (*Lozada et al., 2022*). This microbiome compositional pattern might be caused by a microbial gardening process in which the macroalgae actively stimulates selective microbial growth. However, comparing *U. pinnatifida* biofilm composition in different geographic regions could aid to test the plausibility of this microbial gardening speculation. Furthermore, combining this microbiome biogeographic comparison with host trait measurement (*e.g.*, transcriptomics) could aid to elucidate the extent to which the macroalgae actively selects its epibiotic microbiome and its potential relevance for macroalgae fitness.

Australia is the region where macroalgal holobiont is best understood since it has the highest number of studies (five studies, Table 1) and diversity of molecular approximations (*i.e.*, 16S, shotgun metagenomics, DNA holobiont, Table 1). Yet, most of this research effort has focused only on one species, *Ecklonia radiata* (Laminariales, Lessoniaceae), so it is unlikely that the macroalgal holobiont knowledge is generalizable to all macroalgae hosts in the region (*Marzinelli et al., 2015*, *2018*; *Qiu et al., 2019*; *Song et al., 2021*; *Wood et al., 2022*). Nevertheless, these studies provide valuable insight into the future of macroalgal

holobionts. *Ecklonia radiata* biofilm composition was dysbiotic during environmental stress (*Marzinelli et al., 2015*, *2018*). Yet, stress microbial signatures were not consistent across individuals, which suggests that there are unexplored meaningful covariates (*e.g.*, host genetic variability) that determine the final holobiont phenotype (*i.e.*, biofilm composition) when algae face environmental stress (*Qiu et al., 2019*). Horizontal transfer of genes related to the algae niche specific environment and stress environmental responses between biofilm bacterial members could actively interact to facilitate biofilm adaptation to environmental stress (*Song et al., 2021*). Interestingly, a DNA holobiont approach in the macroalgae *Phyllospora comosa* (Fucales: Seirococcaceae) revealed that host genetic variability had a weak relationship with microbial composition. Moreover, *P. comosa* biofilm composition was driven by local conditions and geography (*Wood et al., 2022*). Together these results suggest a complex interplay in macroalgal holobiont, where genetic variability, biofilm composition, horizontal gene transfer, and environmental conditions are crucial players that create a diverse array of phenotypes. Given the unavoidable environmental changes that will occur in the next few decades and the key role macroalgae play as ecosystem engineers, it is imperative to detect the factors that promote resilience in macroalgal holobiont against environmental stress to maintain the benefits they provide to marine biodiversity.

Although bacterial partners have received most of the attention in algal biofilm research, algal biofilms harbor diverse fungal communities. Fungal biofilm composition has been explored in Antarctica. Fungal epibiotic communities in Antarctic macroalgae are influenced by abiotic (*i.e.*, dissolved oxygen and organic matter) and biotic (*i.e.*, antifungal molecules produced by the host) factors (*Ogaki et al., 2019*). Moreover, macroalgae actively control these communities since some fungal strains associated to algal biofilms exhibit agarolytic and carrageenolytic activity, and can degrade algal biomass (*Furbino et al., 2017*). However, the interactions that take place between bacteria and fungi inhabiting macroalgal biofilm remain unclear. Future studies should have more holistic approaches, where bacteria, fungi, and environmental covariables are simultaneously considered.

In particular, the algal species *M. pyrifera* is distributed worldwide (*i.e.*, present both in the Southern and Northern Hemispheres), and its epibiotic microbial communities have been studied in several regions around the world (*Florez et al., 2019*; *Lin, Lemay & Parfrey, 2018*; *Weigel & Pfister, 2019*). However, little is known about *M. pyrifera* surface biofilm from the Strait of Magellan. In the Strait of Magellan microbiome project, which we will describe at the end of the article, we are attempting to characterize the bacterial and functional traits/responses of *M. pyrifera* in association with environmental factors at different water depths. These data could improve our understanding in macroalgal microbiome response to climate change.

## Invertebrates

### Sponges

Currently, there are 9,542 sponge species around the globe, with at least 8,864 species distributed in the Southern Hemisphere (*de Voogd et al., 2023*; *Downey et al., 2012*). Sponges were the most studied holobionts in the Southern Hemisphere, both in terms of

studied species (65) and number of studies (16), most likely because sponge-associated microbes produce a wide array of metabolites of biotechnological importance (*Taylor et al., 2007*, Table 1). The predominant approach to study sponge holobiont was 16S. However, several studies featured shotgun metagenomic approaches, which shed light on sponge microbiome metabolic potential (Table 1, *Yang et al., 2022*; *Moreno-Pino et al., 2020*, *2021*). Sampling has been performed in all continents from the Southern Hemisphere, but mainly in Western Antarctic Peninsula (Fig. 2).

Sponges are important ecosystem players that participate in several biogeochemical cycles and stabilize benthos (*Bell, 2008*). Sponges are the metazoans that evolved first, and therefore are the sister group to all animals (*Wörheide et al., 2012*). Their anatomy is unlike any other metazoan, but generally consists of several cell layers (*Taylor et al., 2007*). Most of the studies included in our revision used the outermost cell layer. It was explicitly stated in the text when more cell layers were sampled. In a recent study on tropical sponge holobiont conducted in the Northern Hemisphere, there were high metabolic redundancies within the microbiomes that could help buffer sponges from chemical and physical changes in the environment, and from fluctuations in population sizes of individual microbial strains that make up the microbiome (*Kelly et al., 2022*). This is not surprising since their early evolution occurred in a microbe dominated world in the late Precambrian (*Renard et al., 2013*). Sponge physiology seems to be microbe dependent and has developed different symbiotic-based solutions to environmental challenges (*Thomas et al., 2016*).

## Sponge microbiomes in the Southern Hemisphere

Current evidence suggests that sponge microbiome is similar at the phylum level between species in the Southern and Northern Hemispheres (*Taylor et al., 2007*). In the Southern Hemisphere, sponge microbiome is characterized by several core bacterial phyla, including Proteobacteria, Bacteroidetes, Actinobacteria, Verrucomicrobia, Acidobacteria, and Cyanobacteria (*Rodríguez-Marconi et al., 2015*; *Matcher et al., 2017*; *Cárdenas et al., 2018*, *2019*; *Savoca et al., 2019*; *Papale et al., 2020*; *Happel et al., 2022*; *Ruocco et al., 2021*; *Yang et al., 2022*). Interestingly, Thaumarchaeota is the only archaea phylum associated with sponges, yet its association is consistent across several species (*Brochier-Armanet et al., 2008*; *Sacristán-Soriano, Pérez Criado & Avila, 2020*; *Moreno-Pino et al., 2020*; *Steinert et al., 2020*).

Interestingly, some bacterial lineages associated with sponges are phylogenetic novelties, that is, their DNA sequences are new; hence, they could not be identified using current sequence databases. Antarctic (*Papale et al., 2020*; *Moreno-Pino et al., 2021*; *Happel et al., 2022*), Australian (*Yang et al., 2022*), and South African (*Matcher et al., 2017*) sponges had more "unknown" bacterial partners. Interestingly, phylogenetic novelty varied across sponge-associated bacteria genera. For instance, the bacterial genera *Sporosarcina* and *Nesterenkonia* had the greatest phylogenetic novelty. In contrast, bacterial strains from *Cellulophaga algicola* had less phylogenetic novelty (*Moreno-Pino et al., 2021*). Considering metabolic evidence from Northern Hemisphere tropical sponges, functional redundancy in their microbiomes might allow sponges to tolerate environmental

fluctuations, which allows them to be distributed over wide areas (*Kelly et al., 2022*). Hence, environmental variability could be an important driver of microbiota assembly in Southern Hemisphere sponges, at inter- (between different species) and intra-specific (*i.e.*, among individuals of the same species) levels (*Steinert et al., 2020*).

Besides environmental heterogeneity, phylogenetic factors (*i.e.*, species/genera particular affinities) also play an important role in sponge microbiota assembly. For example, sponges from the genus *Mycale* display a strong bacterial core among individuals and species distributed over hundreds of kilometers in the Southern Ocean (*Cárdenas et al., 2018*; *Happel et al., 2022*). Similar stable associations have been found in some Antarctic sponge species (*Steinert et al., 2019*). *Mycale magellanica*, a common sponge living in the Strait of Magellan, shares up to 74% of sequences belonging to Rhodobacteriaceae and Flavobacteriaceae with *Mycale acerate* individuals, a common sponge in the Western Antarctic Peninsula (*Cárdenas et al., 2018*). Moreover, bacterial composition is stable among *M. acerata* individuals distributed across the entire West Antarctic Peninsula (*Happel et al., 2022*).

Similar trends with similar microbiota composition were found in Demospongiae and Hexactinellida sponges from Ross Sea, Antarctica. These sponges share the bacterial genera *Erwinia*, *Methylobacterium*, and *Sphingomonas* (*Papale et al., 2020*). In contrast, in the same region (Ross Sea), sponge microbiota composition has been found to be heavily influenced by environmental microbes (horizontal transmission), both in the cultivable fraction and NGS-microbiota (*Savoca et al., 2019*; *Sacristán-Soriano, Pérez Criado & Avila, 2020*). Overall, this highlights the differences in bacterial composition variability among sympatric sponges, which might create species- or genus-specific microbiota compositions.

Factors underlying microbial recruitment in sponges got more complex when we considered sponge-associated Archaea. For example, in Demospongiae and Hexactinellida sponges from the South Pacific Ocean, bacterial composition was species-specific, while Archaea composition was individual-specific (*Steinert et al., 2020*). Shared Archaea composition was low among individuals, which might suggest that Archaea are opportunistic/contingent players in sponge holobiont, or rather, their functional benefits are widely shared among several archaeal taxa. This pattern contrasted with the stable microbiota composition reported in Ross Sea sponges (*Papale et al., 2020*). The differentiated trends suggest that environmental fluctuation coupled with species-specific filters might drive microbial composition associated with sponges.

The above examples illustrate the complex factors underlying sponge symbiotic associations with bacteria, where horizontal transmission and host-specific factors appear to have a differential role among sponge species. Despite inconsistency among associated microbe identities, it is likely that sponge-associated microbes have similar functional traits. Thus, although sponge microbiota has complex patterns, their microbiomes might have functional convergence (*Cristi et al., 2022*).

Nevertheless, whether sponge microbiome taxonomy obey contingent patterns (*i.e.*, which microbe taxa arrived first or neutral process in microbiota assembly) or indeed have a biological basis is an open question. Symbiotic interactions among sponges and their associated bacteria dynamically shape the sponge microbiota composition. On one hand,

opportunistic bacteria that degrade sponge tissues, such as *Bacillus, Micrococcus*, and *Vibrio*, are common members of the sponge holobiont. On the other hand, there are antibiotic-producing bacteria that regulate the former, such as *Streptomyces, Aquimarina, Pseudovibrio*, and *Pseudoalteromonas* (*Esteves, Cullen & Thomas, 2017*). Variations in quorum sensing, a microbe chemical communication system, might also play an important role in microbial recruitment. The genera *Pseudomonas, Shewanella*, and *Roseobacter*, common bacteria associated to sponges, produce acyl-homoserine-lactone, which is an important chemical messenger in quorum sensing (*Mangano et al., 2018*). The quorum sensing activity by these bacteria might alter the community profile by differential microbial recruitment. Quorum sensing might be an important adaptive trait in sponge holobiont, specially with the ongoing climate-derived marine changes, yet its prevalence, and more important its relevance, among sponge microbiome has not been thoroughly studied.

Metagenomic studies have shed light on the characteristics of the functional repertoire of sponge microbiome. Microbe metabolism differs throughout sponge tissue, highlighting a tissue-specific microbiome metabolism (*Yang et al., 2022*). Several examples pointed to nutrient provisioning as an important trait in sponge microbiome. Microbial symbionts encode multiple genes related to nitrogen fixation and metabolism of nitrogen compounds, sugars derived from photosynthesis (*Moreno-Pino et al., 2020*), as well as vitamin B5 (*Moreno-Pino et al., 2021*). In consonance with antagonistic interactions among members of the sponge microbiome, antibiotic resistance and biopolymer degradation (*Moreno-Pino et al., 2021*), as well as CRISPR genes, transposases, detoxification genes, and restriction site modifications (*Moreno-Pino et al., 2020*) are common traits in the sponge microbiome. The latter functions highlight the evolution of the microbiome within the sponge itself since several microbes associated with sponges degrade the sponge tissues and are unaffected by antimicrobial compounds.

Furthermore, the high prevalence of CRISPR genes in the sponge microbiome suggest that their bacterial members are under constant phage attack (*Moreno-Pino et al., 2020*), which adds another complexity layer to the microbial interactions in the sponge microbiome. There was a high proportion of genes in the microbial communities with unknown functions, so besides the phylogenetical novelty described above, functional/ metabolic microbial novelty associated with sponges also stood out (*Moreno-Pino et al., 2020*).

To date, few studies have addressed the negative effects that marine climate change will have on sponge holobiont. An interesting exception is *Kandler et al. (2018)*, who found that microbial communities of tropical sponges from New Guinea, *Coelocarteria singaporensis* and *Stylissa* cf. *flabelliformi*, might be tolerant to future marine pH conditions. However, it is still a unifactorial experimental approach that does not represent a reliable test of the multifactorial climate change process. In contrast, there are other climate change-driven effects, such as ice scour (seabed modification caused by floating icebergs), which is predicted to increase as a direct consequence of sea surface temperature rise. Hence, ice scour might pose a major challenge to sponges in polar environments. Ice scour damages benthic communities, especially sponges, like *Isodictya kerguelenensis.*

Ice scour injuries produce microbial fingerprints that are easily identified (*Rondon et al., 2020*). Thus, as climate change progresses and ice scour increases, Antarctic sponges integrity might be compromised in the next few decades.

## Non-sponge invertebrate microbiomes in the Southern Hemisphere

Marine invertebrates comprise between 35–39 recognized phyla (*Valentine, 2006*; *Zhang, 2013*). The most iconic marine invertebrate phyla are Mollusca (118,061 species), Echinodermata (20,550 species), Annelida (17,426 species), Cnidaria (17,426 species), and Bryozoa (11,474 species) (*Zhang, 2013*). Marine invertebrate microbiomes have received little attention with 23 species within 10 research articles (Table 1). Interestingly, this group has outstanding examples of authentic holobiont approaches (*i.e.*, coupled measurement of host and microbe traits) (Table 1). Sampling has been conducted mainly in Australia, Antarctica, and Africa. Notably, there were no marine invertebrate microbiome studies in South America (Fig. 2).

Overall, the main bacterial phyla in marine invertebrate microbiome were Proteobacteria, Bacteroidetes, Verrucomicrobia, Tenericutes, and Actinobacteria (*Webster & Bourne, 2007*; *Murray et al., 2016*, *2020*; *Unzueta-Martínez et al., 2022*). Antarctic corals and snails had stable microbiota among individuals. The Antarctic soft coral *Alcyonium antarcticum* has a core microbiota composed of Proteobacteria, Bacteroidetes, Firmicutes, Actinomycetales, Planctomycetes, Chlorobi, and sulfate-reducing bacteria (*Webster & Bourne, 2007*). A similar trend was observed in the Antarctic snail *Synoicum adareanum*, whose microbiota was characterized by a high prevalence of Proteobacteria, Verrucomicrobia, Actinobacteria, Nitrospirae, and Bacteroidetes (*Murray et al., 2020*). In other cases, microbes associated to invertebrates displayed low diversity, such as the ice-adhered anemone *Edwardsiella andrillae*, endemic to the Ross Sea, whose main phyla were Proteobacteria and Tenericutes. Interestingly, most of its sequences showed recent diversification branching, which suggests that its associated bacteria are evolutionarily recent (phylogenetically new) (*Murray et al., 2016*).

Invertebrates with long and complex life cycles also had a complicated pattern in their associated microbes along their life stages. In the Sydney rock oyster (*Saccostrea glomerata*), bacterial composition was driven by life history characteristics (*Unzueta-Martínez et al., 2022*). For example, environmental bacteria were a major source of bacterial composition in swimming larvae stages, thus, the microbiota of these stages was characterized by common marine free-living bacteria. On the contrary, sessile stages, such as pre adult stage, adult and gametes had distinct microbiota profiles. Overall, the bacterial composition across life stages in the Sydney rock oyster varied, which suggests that most oyster-associated microbes are either opportunistic or commensals, with little relevance to the oyster. Nevertheless, the genus *Nautella* (Rhodobacterales) was consistently present across stages, and its abundance notably increased in the last stages.

## Crustacean microbiomes in the Southern Hemisphere

To the best of our knowledge, the only crustaceans in the Southern Hemisphere that have been studied to date are lobsters, krill, and copepods. Overall, the bacterial microbiota of

crustaceans in the Southern Hemisphere was dominated by Campilobacterota, Tenericutes, Actinobacteria, Firmicutes, Bacteroidetes, and Proteobacteria (*Clarke et al., 2019*; *Ooi et al., 2019*; *Clarke et al., 2021*; *Oh et al., 2021*; *Zhang et al., 2021*).

Microcrustaceans are important trophic links between primary producers (*e.g.*, diatoms) and predators (*e.g.*, seabirds and fish). Despite its inherent exposure to marine bacteria, Antarctic krill hosted unique bacteria phyla in its body; furthermore, its epibiotic-associated bacteria differentiated as geographic distance increased. Hence, distance, rather than environmental heterogeneity drove epibiotic bacteria composition in krill (*Clarke et al., 2019*; *Clarke et al., 2021*). The major bacterial players on krill chitin surface were Campilobacterota and Tenericutes, and Actinobacteria and Firmicutes in the stomach and intestinal gland (*Clarke et al., 2019*). Interestingly, *Colwellia* bacteria was a prevalent member of epibiotic microbiota in Antarctic krill swarms on local and regional scales (*Clarke et al., 2021*). Its persistent association, over thousands of kilometers, might suggest an important role in krill health.

Sea temperature increase might disrupt psychrophilic bacteria associated with Antarctic crustaceans. In the Antarctic copepod *Tigriopus kingsejongensis*, 15 °C temperature treatment had profound effects on its fecal microbiome. Temperature increase diminished abundance of psychrophilic bacteria (*e.g.*, *Colwellia*) but facilitated the increase of opportunistic pathogens (*e.g.*, *Vibrio*) and virulence genes (*Oh et al., 2021*). As sea surface temperature increases, it is probable that microcrustacean microbiomes from polar latitudes of the Southern Hemisphere will face major shifts in its associated bacteria. Bacteria that prefer higher temperatures (*e.g.*, *Vibrio*) will increase, while psychrophilic bacteria will decrease.

Natural life history traits, such as molting and juvenile susceptibility to temperature increase, had important effects on crustacean microbiota composition (*Ooi et al., 2019*; *Zhang et al., 2021*). Molting is a critical process in crustaceans for growth and sexual maturation. Evidence from the Chinese mud crab (*Scylla paramamosain*) suggests that molting represents a bottleneck to most of its associated microbes in gills and midgut. Nevertheless, hemolymph bacteria *Halomonas* and *Shewanella* prevailed despite molting (*Zhang et al., 2021*). Noteworthy, the abundance of these bacteria is highly correlated with the crab antimicrobial gene expression. These results suggest the presence of highly adapted bacteria to the complex life cycle of this mud crab. Yet, it is uncertain to what degree molting could alter microbiome traits in crab holobiont. Sea temperature increase might impose microbe-related burdens to crustaceans, as exemplified by the spiny lobster *Panulirus ornatus* in Australia (*Ooi et al., 2019*). In this spiny lobster, temperature increase had a direct relationship with juvenile mortality. As temperature increased, so did bacteria metabolism, which burdened the lobster immune system by bacteria infiltration and subsequent uncontrolled proliferation in the hemolymph.

Interestingly, holobiont approaches have been conducted in snails in deep hydrothermal vents and several Lucinidae species (Mollusca) across the world, where host and microbial DNA have been simultaneously studied (*Lan et al., 2021*; *Osvatic et al., 2023*). These results have highlighted the effect of ecological niche and host-microbe metabolic complementarity in microbiome assembly. Interestingly, sulfur-oxidizing

bacteria were present in phylogenetical and geographical distant species of Lucinidae Mollusca (*Lan et al., 2021*; *Osvatic et al., 2023*). The above examples highlight the relevance of coupling holobiont approaches with relevant ecological data to address the meaningfulness of the interactions among hosts and their associated microbes.

In the Strait of Magellan study that we will describe at the end of the article, we are including two crustacean species, the centolla (*Lithodes santolla*) and channel sprawns (*Munida gregaria*). The former, is an economically important species that spends most of its life on the sea floor, while the latter is a key species in trophic energy transfer since it is simultaneously an important plankton consumer and is eaten by several predators (*e.g.*, Magellanic penguins and sea lions). In our holobiont study in the Magallanes region, we expect that the centolla will present fewer signs related to UV light and heat stress than any of the other species that have a more ample niche in the water column or on the surface, as is the case of sea lions and penguins.

## VERTEBRATES

### Fish microbiomes in the Southern Hemisphere

There are more than 20,000 species of marine fishes around the globe (*Census of Marine Life, 2003*).

Fishes are important links between trophic basal and higher levels. However, the fish holobiont was the least studied in the Southern Hemisphere with nine studied species within four research articles (Table 1). The predominant molecular approach to study the marine fish microbiome has been 16S metabarcoding (Table 1). Fish microbiome sampling effort in the Southern Hemisphere has been predominantly performed in temperate (Australia) and Antarctic latitudes (Antarctica) (Fig 2). Hence, there are important gaps in tropical, temperate, and cold latitudes, specifically in South America, Africa, and Indonesia (Fig. 2). Importantly, we still ignore the microbiome of native fish species since most studies were conducted in commercially important species. Most fish microbiome studies have aimed to test the usefulness of microbial taxa as biosensors of fish health. Fish skin has a mucous layer over its epidermis that serves as an additional barrier between the environment and the host skin. The mucous layer consists of immunogenic compounds that play important roles in innate and adaptive immunity (*Gomez, Sunyer & Salinas, 2013*). Thus, bacteria inhabiting fish skin might be commensals in healthy individuals or opportunistic/pathogenic in unhealthy fish.

Current evidence of fish bacterial microbiota in the Southern Hemisphere has shown that the most prevalent bacteria phyla are Actinobacteria, Firmicutes, Proteobacteria, Tenericutes, and Bacteroidetes (*Song et al., 2016*; *Minich et al., 2020*; *Legrand et al., 2018*; *Heindler et al., 2018*). In the Southern bluefin tuna (*Thunnus maccoyii*) from Portland Australia, captivity and antiparasitic treatment (*i.e.*, praziquantel) have important effects on fish microbiota composition. Healthy fish microbiota, without praziquantel, is dominated by Mycoplasmataceae on the skin and *Pseudomonas*, *Acinetobacter*, *Brevundimonas*, and *Delfita* in the gut (*Minich et al., 2020*). In the yellowtail kingfish (*Seriola lalandi*), in temperate and southern waters of Australia, early enteritis produces microbial fingerprints in skin microbiota. Early enteritis is associated with greater

abundance of *Loktanella, Marivita, Planktomarina, Simplicispira*, and *Litoricola*, as well as decreased diversity in the microbial community (*Legrand et al., 2018*).

Fish microbiota has also been considered under a natural history framework. In four species of wild Antarctic fish (*Trematomus bernacchii* (family Nototheniioidei), *Chionodraco hamatus* (family Channichthyidae), *Gymnodraco acuticeps* (family Bathydraconidae), and *Pagothenia borchgrevinki* (family Nototheniioidei)), gut microbiota had a stable composition among several species (up to 50% sequences were shared among individuals) (*Song et al., 2016*). This suggests the presence of a core intestinal microbiome in Antarctic fish despite differences in environment and diet, which might play important roles in fish health.

Interestingly, fish gut microbiota could serve as a biological prognosis of anthropogenic impact in marine environments by comparing historical and contemporary samples. In the Antarctic fish *Trematomus* spp., historical samples (museum samples over 100 years old, fixed with formalin and embedded in paraffin) have revealed notable shifts in gut microbiota composition. Contemporary fish gut microbiota was characterized by Chlamydia, Firmicutes, Cyanobacteria, and Mycoplasma. In contrast, historical fish gut microbiota was dominated by Proteobacteria. Despite the richer phyla in contemporary fish, OTU richness and Shannon index diversities were higher in ancient fish (*Heindler et al., 2018*). These results attempt to elucidate the relationship between fishing practices and fish gut microbiota, in a historical context. Before global fishing practices, fish were able to have a consistent diet that produced redundant gut communities at the phylum level. In contrast, fishing practices have disrupted prey availability, which has forced fish to become more opportunistic in their feeding, producing gut communities with wider phylogenetic representation, albeit less diversity (*Heindler et al., 2018*). Whether the shift from more diverse gut microbiota represented by one phylum to less diverse communities with members spanning several phyla has impacted fish (and any marine host) health, is a deep open question.

In the Strait of Magellan project that we describe at the end of the article, we study two fish species, the sardine *Sprattus fuegensis* and the farm salmon *Salmo salar*. We think that the skin microbiome comparison between a wild native species (*i.e.*, sardine) and an introduced species raised in captivity (*i.e.*, salmon) will provide important clues about local marine conditions as well as the effect of captivity.

## Seabird microbiomes in the Southern Hemisphere

Seabirds are essential for ecosystem stability in the Southern Hemisphere (*Woehler et al., 2001*). Currently, there are 350 species worldwide, with at least 61 species endemic to the Southern Hemisphere (*Croxall et al., 2012*). Seabird microbiome has received substantial attention in the Southern Hemisphere, with 22 studied species in 16 research articles (Table 1). Nevertheless, most of these studies were restricted to a few penguin species (Table 1) and conducted using fecal samples (but see *Leclaire et al., 2019*). However, this is a global trend, since in the Northern Hemisphere there were few studies addresing feather bacterial communities, most likely because these are challenging species to sample (*Pearce et al., 2017*; *Leclaire et al., 2019*). Seabird microbiome sampling effort in the Southern
Hemisphere has been performed in most latitudinal regions (temperate (Brazil & Australia), cold (Argentina and Keguelen) and Antarctic (Antarctic islands and Western Antarctic Peninsula), however there were no studies in tropical latitudes (Fig. 2). The seabird microbiome has been predominantly studied using taxonomic marker approaches (*i.e.*, 16S); hence associated microbes have been predominantly addressed at the bacterial community level. However, there are metatranscriptomic and RNA viromic surveys (*Smeele, Ainley & Varsani, 2018*; *Marcelino et al., 2019*; *Wille et al., 2020*).

Overall, the main bacterial phyla in penguin fecal microbiota from the Southern Hemisphere were Firmicutes, Proteobacteria, Actinobacteria, and Bacteroidetes (*Potti et al., 2002*; *Barbosa et al., 2016*; *Dewar et al., 2013*, *2014*, *2017*; *Yew et al., 2017*; *Lee et al., 2019*; *Tian et al., 2021a*, *2021b*). In turn, the main bacterial phyla in seabird plumage microbiota from the Southern Hemisphere were Actinobacteria, Proteobacteria, Firmicutes, and Acidobacteria (*Leclaire et al., 2019*).

Aging is an important driver of fecal microbiota composition, both in wild and captive penguins (*Barbosa et al., 2016*; *Dewar et al., 2017*; *Tian et al., 2021a*). In wild chinstrap penguin chicks (*Pygoscelis antarctica*), fecal microbiota is dominated by Firmicutes, specially by Clostridiales, *Leuconostoc*, and *Fusobacterium*. In contrast, adult fecal microbiota was dominated by Proteobacteria and Bacteroidetes, specially by Neisserales, Fusobacteriales, and Campylobacteriales, yet there was high variability among individuals (*Barbosa et al., 2016*; *Lee et al., 2019*). In contrast, in captive chinstrap penguins at the Dalian Sun Asia Aquarium, China, aging only changed relative abundances of the main constituents of fecal microbiota. Chick fecal microbiota was dominated by *Acinetobacter*, while *Pasteurella* was the dominant bacteria in senior penguins (between 22—28 years old); the fecal composition in chicks and adult penguins included *Clostridium* and *Fusobacterium* (*Tian et al., 2021a*). Furthermore, these compositional shifts followed predicted functionality shifts. In general, predicted functionality reached its maximum diversity in adults, while it started to decline in senior penguins (over 22 years old) (*Tian et al., 2021a*).

Host age importance in fecal microbiota composition was confirmed in little blue penguins (*Eudyptula minor*) at the Phillip Island Nature Parks, Australia, where fecal microbiota between chicks and adults differed (*Dewar et al., 2017*). Differences in fecal microbiota composition might be explained by the kind of food chicks and adults eat. Chicks eat regurgitated food, which might not require a robust microbial metabolic repertoire to aid in digestion; in contrast, adults eat raw food that might contain recalcitrant chemicals, such as domoic acid in fish (*Lefebvre et al., 2002*) or fluoride, from krill (*Yoshitomi & Nagano, 2012*).

Besides community compositional comparisons, 16S surveys have been used to understand changes in penguin fecal microbiome predicted metabolic functions. For example, in captive gentoo penguins (*Pygoscelis papua*) at the Dalian Sun Asia Aquarium, China, sex apparently influenced fecal microbiome predicted metabolism (*Tian et al., 2021b*). Male fecal microbiota was enriched in carbohydrate metabolism, putatively driven by the Lachnospiraceae family, whereas female fecal microbiota was enriched in protein metabolism, putatively driven by the Fusobacteriaceae family (*Tian et al., 2021b*).

Microbiome-predicted functions from 16S data are constrained by the number of available microbial genomes sequenced (*Douglas et al., 2020*) and information on "optimal performance" is limited to human samples and decrease sharply in environmental samples (*Sun, Jones & Fodor, 2020*). Therefore, functional microbiome studies (*i.e.*, metagenomics or metatranscriptomics) coupled with experimental/culture assays are needed to validate the metabolic functions that have been attributed to penguin fecal microbiota.

Fecal microbiota comparisons among several penguin species have shed light on penguin species-specific factors influencing fecal microbiota composition. For example, *Dewar et al. (2013)* compared the fecal microbiota composition of four penguin species: macaroni penguins (*Eudyptes chrysolophus*), King penguins (*Aptenodytes patagonicus*), gentoo penguins from two sites (Bird Island in South Georgia and Baie du Marine, Possession Island Crozet Archipelago), and little blue penguins from Phillip Island, Australia. This study revealed interesting patterns in penguin fecal microbiota at phyla level, where the dominant bacterial phylum in each penguin species was as follows: Firmicutes in macaroni penguins, Actinobacteria in King penguins, and Proteobacteria in gentoo and little blue penguins (*Dewar et al., 2013*). Fecal microbiota divergence among penguin species might be explained by trophic niche differences. Additionally, hormone profiles might also have specific effects on gut microbiota composition. Differences could also be attributed to geographical factors. Systematic studies, including multiple colonies of each species with an adequate sample size, are needed to verify to what extent these results reflect penguin fecal microbiota in these species.

Penguin gastrointestinal tract is heterogeneous along its structure. Given that fecal microbiota represents the last section of the gastrointestinal tract, it is unlikely that it is a representative sample of penguin gastrointestinal microbial diversity. Indeed, stomach microbiota studies conducted in Adélie (*Pygoscelis adeliae*) and chinstrap penguins from Signy Island, South Orkney Islands, Antarctica revealed differences in bacterial community composition between the stomach and fecal microbiota. The stomach microbiota of Adélie and chinstrap penguins was characterized at the phyla level by Firmicutes, Proteobacteria, Fusobacteriota, and Tenericutes. Common genera in both species were: *Cetobacterium, Psychrobacter, Chelonobacter, Clostridium* (family: Clostridiaceae), *Mycoplasma*, and *Ornithobacterium* (*Yew et al., 2017*). The stomach bacterial communities differed from those reported in the fecal microbiota of these species, where Actinobacteria and Firmicutes were the dominant phyla in Adélie penguin fecal microbiota (*Banks, Cary & Hogg, 2009*), while chinstrap penguin fecal microbiota was characterized by Proteobacteria and Bacteroidetes (*Barbosa et al., 2016*; *Tian et al., 2021a*). Stomach bacterial differentiation might be explained in part by the presence of spheniscins, special compounds in the penguin stomach that prevent bacteria from digesting food (*Thouzeau et al., 2003*).

Fasting is an important stage in penguins' life history. Penguins fast when they rear their chicks and when they molt. Fasting produces species-specific compositional changes in penguin fecal microbiota (*Dewar et al., 2014*; *Lee et al., 2019*). This could be explained by differences in fast length among penguin species and their fecal-associated microbes. For instance, in King penguin, fasting increased the relative abundance of Proteobacteria,

Firmicutes, Actinobacteria, *Fusobacteria*, and *Bacteroidete*s. In contrast, in little blue penguin, fasting decreased most phyla relative abundance (*Dewar et al., 2014*). *Fusobacteria* is a butyrate-producing bacterium. Butyrate is a known anti-inflammatory agent (*Canani et al., 2011*). Furthermore, butyrate administration in chickens improves immune system, which improves health and decreases pathogen incidence (*Panda et al., 2009*). The enrichment of *Fusobacteria* in fasting King penguins suggests that this bacterium may play an important role in metabolic homeostasis. Further evidence supporting the effect of fasting in penguins came from gentoo and chinstrap penguins in different trophic status (*i.e.*, feeding season or fasting). In feeding chinstrap and gentoo penguins, Fusobacteriota and Proteobacteria were the dominant bacteria, while in molting (fasting) birds, these phyla decreased while Firmicutes relative abundance increased. Nevertheless, the magnitude in the compositional changes of fecal microbiota composition differed among species; while shifts in chinstrap penguins were subtle (*i.e.*, not supported statistically), those in gentoo penguin were major (*i.e.*, statistically supported) (*Lee et al., 2019*).

It is worth mentioning that penguin and Antarctic bird fecal microbiota studies have shed light on the widespread occurrence of genera related to known pathogens. Those genera include *Campylobacter, Yersinia, Salmonella*, and *Escherichia* (*Barbosa & Palacios, 2009*), yet it is uncertain to what degree they affect penguin health, but they might have a parasitic basis. This point was supported by studies in Magellanic penguin chicks from Peninsula Valdez, Argentina whose fecal microbiota was dominated by *Corynebacterium*, which appears to divert resources from chicks and hinder their growth. Interestingly, administration of a wide spectrum antibiotic reduced *Corynebacterium* abundance, reversing growth halt (*Potti et al., 2002*).

However, the presence of potential pathogens in Antarctic animals can be a misinterpretation of what "is normal" and anthropogenic (*Souza et al., 1999*). The same may occur with viruses associated with Antarctic fauna. Antarctic seabird fauna, such as gentoo, chinstrap, Adélie penguins, rockhopper (*Eudyptes chrysocome*), and south polar skua (*Stercorarius maccormicki*) harbor diverse viral communities (*Smeele, Ainley & Varsani, 2018*). In addition, Antarctic penguins (chinstrap, Adélie, and gentoo) harbor a great diversity of viruses in their cloaca and are in contact with diverse viral communities from their ectoparasitic mites (*Wille et al., 2020*). These results highlight the uniqueness of fauna living in the Antarctic biome. Moreover, it calls for further microbiome research in these remote places to elucidate their relationship with worldwide fauna.

Seabird plumage microbiota has been poorly studied in the Southern Hemisphere, yet we got some insights from the blue petrel (*Halobaena caerulea*), whose plumage microbiota composition was highly variable among body sites (*Leclaire et al., 2019*). Furthermore, some bacteria showed a positive correlation with the major histocompatibility index (MHC), which suggests that plumage bacteria are influenced by MHC allele diversity in this seabird.

Interestingly, a metatranscriptomic approach revealed a high incidence of antibiotic resistant genes in seabirds from Australia, which differed based on their trophic ecology (*Marcelino et al., 2019*). Synanthropic (living near human settlement) species with filter

eating habits, such as Australian ducks (*Anas* spp. and *Tadorna tadornoides*), had the highest diversity of antibiotic resistant genes. In contrast, avocets (*Recurvirostra novaehollandiae*) and gentoo penguins, which live in remote areas and actively prey for food, had the fewest. However, the presence of antibiotic resistant genes is not a surprise since this may a very ancient strategy in microbial communities (*Souza et al., 1999*). Interestingly, even though the cloacal microbiome of gentoo penguins had the lowest antibiotic resistant gene diversity, it displayed resistance against unique drugs, such as macrolides, lincosamides, and streptogramins (*Marcelino et al., 2019*).

In a similar fashion, kelp gulls (*Larus dominicanus*) and Magellanic penguins from Brazil were assessed using qPCR to evaluate the diversity of antibiotic genes they harbor (*Ewbank et al., 2021*). Ecological strategies (synanthropic/remote & migratory/non opportunistic/specialized feeding) might have a strong association with antibiotic gene resistance transmission, with those related to anthropocentric activities having the greatest diversity in antibiotic resistance genes. As expected, kelp gull, a synanthropic species, had the greatest antibiotic resistance gene diversity. Its antibiotic resistant gene pool was resistant against eight drugs: tetracycline, aminoglycosides, sulfonamides, chloramphenicols, macrolides, quinolones, betalactams, and polymyxins. On the contrary, in the Magellanic penguin, a migratory, non-synanthropic, specialized feeder (it preys mainly on fish and squid), antibiotic resistance gene diversity was lower, with specific resistance against two drugs, tetracycline and quinolones (*Ewbank et al., 2021*).

In our project on the surface microbiome of key species in the Strait of Magellan, we are planning to study the feather microbiome in Magellanic and King penguins. We are interested in testing geographical, phenological, and developmental effects on feather microbiome of these species. These studies will provide original baseline knowledge regarding penguin feather associated bacteria.

## Marine mammal microbiomes in the Southern Hemisphere

Marine mammals comprise several species that are collectively regarded as "marine sentinels" since their population trends could give valuable insights about the status of the marine ecosystem (*Moore, 2008*). Currently, there are 115 species distributed around the globe (*Kaschner et al., 2011*). Up to one third of them cooccur between the temperate —cold latitudes (*i.e.*, 20—50°) of the Southern Hemisphere. Important marine mammal biodiversity hotspots include New Zealand, Sub-Antarctic and Southeastern Pacific islands, and offshore waters along the coasts of southern South America (*Kaschner et al., 2011*). Marine mammal microbiome has received the greatest attention in the Southern Hemisphere, with 15 studied species in 17 research articles (Table 1). Nevertheless, most studies have been conducted in the humpback whale (Table 1). In most cases, these studies were conducted using 16S approaches, but there was a pinniped shotgun metagenome (Table 1, *Smith et al., 2013*). Interestingly, marine mammal microbiome research has been done in all continents from the Southern Hemisphere, with sampling effort across all latitude regimes (tropical, temperate, cold, and Antarctic). In terms of number of studies, marine mammals from temperate latitudes (*i.e.*, Australia) have been the most studied, followed by Antarctic and cold latitudes (*i.e.*, South America) (Fig. 2).

Cetacean microbiomes have been studied in several species from the Northern Hemisphere (*Sanders et al., 2015*; *Van Cise et al., 2020*; *Apprill et al., 2020*; *Miller et al., 2020*). However, we still lack a more comprehensive view of cetacean microbiome from the Southern Hemisphere since most of the current studies have been conducted in one species over different geographic locations (*i.e.*, humpback whale) (Table 1). While this gives us a deep understanding of humpback whale microbiome across geographic regions, it also pinpoints the great void that remains in other cetaceans, such as whales and dolphins. Besides humpback whale, several studies have been conducted in pinnipeds (Table 1). Marine mammal microbiome research has focused on whale skin and blow microbiota studies. In the Southern Hemisphere, skin microbiological surveys have been done in rorquals (Balaenopteridae, species *Megaptera novaeangliae* (humpback whale), *Balaenoptera musculus*, and *Balaenoptera physalus*) as well as killer whales (*Orcinus orca*). Overall, the main bacterial elements in whale skin microbiota were Proteobacteria, Bacteroidetes, Actinobacteria, and Firmicutes (*Apprill et al., 2014*; *Pirotta et al., 2017*; *Bierlich et al., 2018*; *Hooper et al., 2019*; *Vendl et al., 2019*; *Toro et al., 2021*).

Humpback whales are the best studied cetacean in the Southern Hemisphere. Studies on humpback whales from the Samoa islands (South Pacific), Chilean coasts, and Antarctic regions have allowed the detection of a skin core microbiota, characterized by *Tenacibaculum* and *Psychrobacter* (*Apprill et al., 2014*; *Bierlich et al., 2018*; *Toro et al., 2021*). Rorqual (*M. novaeangliae*, *B. musculus*, and *B. physalus*) skin microbiota along several points in Chilean coasts had idiosyncratic and species-specific trends, that is, they had a unique skin microbial composition at the individual and the interspecific level. However, their skin microbiota alpha diversity was similar in compositional terms (*i.e.*, Shannon diversity), but was slightly different in phylogenetic terms (*i.e.*, Faith phylogenetic diversity) (*Toro et al., 2021*). Skin microbiota of humpback whales foraging in Antarctica and the Strait of Magellan were enriched in *Psychrobacter* bacteria. Changes in sea surface temperature and shifts towards northern areas were associated with decreases in *Psychrobacter* relative abundance. This pattern suggests that sea temperature is an important driver in humpback whale skin microbiota assembly (*Bierlich et al., 2018*; *Toro et al., 2021*).

Blow microbiota of Australian humpback whales has been surveyed to address whale health. In contrast with skin microbiota, blow microbiota was sparse among individuals, without any discernable core. The most abundant microbes in the blow were *Tenacibaculum, Pseudomonas, Leptotrichia*, and *Corynebacteria*. Additionally, some individuals had potential respiratory pathogens in their blow microbiota, such as *Balneatrix, Clostridia, Bacilli, Staphylococcus*, and *Streptococcus* (*Pirotta et al., 2017*; *Vendl et al., 2019*). Furthermore, whale blow-associated bacteria harbor significant phylogenetic novelty since half of the sequences in some individuals were only identifiable at the class level (*Vendl et al., 2019*). Whale feeding phenology is an important factor that could underlie the sparsity of blow microbiota structure. While feeding, blow core microbiota was composed of *Arcobacter, Corynebacterium, Enhydrobacter, Helcococcus*, and *Tenacibaculum*, albeit in very low abundance (1.5% or less). In contrast, during migration, when whales were fasting, blow microbiota among individuals became highly variable,

with no discernible core (*Vendl et al., 2020a*). Besides phenology or health status, whale blow microbiota was also influenced by the sociality degree of the studied species. Whale species with gregarious habits (*e.g.*, humpback whales) have higher diversity and a great microbial core in their blow microbiota, in contrast with more solitary whales (*Vendl et al., 2020b*). This likely reflects a horizontal transmission of blow microbes among contiguous whales, where the microbes exhaled by one individual are inhaled by another, and so on. Nevertheless, more studies are needed to determine whether this pattern emerges because of horizontal transmission or reflects a common health status among whale groups.

In the case of killer whales, the Antarctic ecotypes have been found to harbor a distinct skin microbiota from ecotypes in the Northern Hemisphere. Differences were driven by *Tenacibaculum dicentrarchi* bacteria, diatoms, and several algae-associated bacteria (*Hooper et al., 2019*). More systematic studies of whale-associated bacteria and ideally "authentic" holobiont approaches (*i.e.*, describing whale genetic traits, as well as their microbiome and trancriptome) are needed to better understand the whale holobiont.

In the project on the surface microbiome of key species in the Strait of Magellan (*ANID R20F0009, 2020*), we are including a holobiont approach to studying the humpback whales of the area. In particular, we are sampling individuals that migrate to the Strait of Magellan to feed in the austral summer season.

Pinnipeds are apex predators, whose health might provide information about marine ecosystem conditions (*Moore, 2008*). Sampling pinnipeds is challenging due to the remote location of their colonies and proclivity to escape from humans; hence, the most feasible samples to study them are the feces they leave on rocks. Seal fecal microbiota is apparently driven by several host factors, including feeding, geographic distribution, ontogeny, trophic niche differences, anatomy, and physiology. This may explain, at least in part, the great fecal microbiota composition variability among species where there is no detectable fecal core microbiota. Nevertheless, it is uncertain whether the taxonomic differences are congruent with microbiome metabolic traits.

Overall, the main bacterial phyla in seal fecal microbiota were Firmicutes, Fusobacteria, Bacteroidetes, Proteobacteria, and Actinobacteria (*Nelson et al., 2013*; *Delport et al., 2016*; *Grosser et al., 2019*; *Kim, Cho & Lee, 2020*; *Toro-Valdivieso et al., 2021*). Nevertheless, studying pinniped fecal microbiota has been challenging as there seems to be a high proportion of phylogenetic novelty that interferes with taxonomic identification (*i.e.*, at the family or genus level) and proper analysis of the divergence among communities (*Toro-Valdivieso et al., 2021*). For example, fecal microbiota composition at phylum level seemed to be identical between pinniped species (*Nelson et al., 2013*; *Kim, Cho & Lee, 2020*) but had species-specific compositional patterns at lower taxonomic levels. For instance, in the southern elephant (*Mirounga leonina*) and Weddell (*Leptonychotes weddelli*) seals fecal microbiota at the phylum level was dominated by Firmicutes, yet at the family level Ruminococcaceae and Acidaminococcaceae, respectively, drove differences in the fecal microbiota composition (*Kim, Cho & Lee, 2020*). Life history characteristics created complex patterns that influenced fecal microbiota, as shown with the fecal microbiota of southern elephant and leopard (*Hydrurga leptonyx*) seals. In these species, fecal microbiota was shaped by the simultaneous effect of species, age, and sex, creating complex patterns

with no discernable trend (*Nelson et al., 2013*). In the Australian fur seal (*Arctocephalus pusillus*), aging from pups to adults produced a successional pattern in fecal microbiota composition. Adult fecal microbiota had unique bacterial taxa dominated by *Clostridium, Lactobacillus*, and *Enteroccocus*. The diet shift from milk with high fat-protein in pups to a marine raw diet in adults is thought to underlie the fecal microbiota diversification (*Smith et al., 2013*).

A fecal microbiota survey in the Australian sea lion (*Neophoca cinerea*) showed a core fecal microbiota at the family level, composed of Clostridiaceae bacteria. This core only arose in wild seals, which suggests that natural diet might represent a cohesive driver in fecal microbiota composition. Meanwhile, captive animals lacked Clostridiaceae in their fecal microbiota. Moreover, seals from wild colonies had more fecal microbial diversity than captive colonies, especially those with high densities (*Delport et al., 2016*). At the functional level, the Australian sea lion fecal microbiome was enriched in carbohydrate metabolism, nitrogen biosynthetic and nutrient transport pathways, as well as virulence genes. The fecal microbiome composition appeared to play an important role in fat—nutrient storage, crucial to marine mammal survival in polar ecosystems (*Lavery et al., 2012*). This could explain a fat storage mechanism driven or highly influenced by gut microbes. Nevertheless, a higher sampling number and more phylogenetic inclusivity (*i.e.*, more seal and sea lion species) are needed to validate the prevalence of these microbiome traits among populations and pinniped species. Interestingly, Clostridiaceae had a high prevalence among Australian seal species fecal microbiota (*Lavery et al., 2012*; *Smith et al., 2013*; *Delport et al., 2016*), which might suggest that they play an important role in seals digestion. Nevertheless, it is a cautionary interpretation since gut microbiota composition varies along the gastrointestinal tract. Hence, fecal microbiota reflects only a portion of the gut microbiome.

In the case of skin microbiota, there was one example from the Antarctic fur seal (*Arctocephalus gazella*) (*Grosser et al., 2019*). In this fur seal, colony density drove skin microbiota structure, rather than genetic similarity. High density colonies had lower alpha diversities in their skin microbiota. This suggests that stress associated with overcrowding could affect skin microbiota composition, diminishing the richness and abundance of bacterial taxa. Nevertheless, overcrowding might also facilitate horizontal transmission since fur seals were in close contact. This could allow the fast transmission of opportunistic microbes capable of dominating the community. The combined explanations might explain the pattern: stress produced by overcrowding, coupled with increased transmission of fast reproducing microbes shaped the skin microbiota of colonies with high density (*Grosser et al., 2019*).

Interestingly, several viral surveys have been conducted in Antarctic seals, particularly in the elephant and Weddell seals (*Smeele, Ainley & Varsani, 2018*). These surveys have highlighted the diverse viral communities that Weddell seals harbor, namely polyomavirus, anelloviruses, and alphaviruses (*Tryland et al., 2005*; *Varsani et al., 2017*; *Fahsbender et al., 2017*). In elephant seals, viral communities were characterized by arboviruses and alphaviruses (*La Linn et al., 2001*; *Forrester et al., 2012*). These studies have highlighted the importance of viral surveys in Antarctic fauna. Moreover, they

encourage the implementation of viral surveys in more vertebrate hosts across the Southern Hemisphere so we could better grasp viral reservoirs in wildlife and the probability of encountering new pathogens of human importance.

In the Strait of Magellan project (ANID R20F0009, 2020), we are studying the skin surface microbiome in colonies of the South American sea lion (*Otaria byronia*), a keystone species in the sub-Antarctic ecosystem.

## Microbiomes in the southern ecosystems and climate change

In the southern ecosystems, including Antarctica, it has been predicted that higher temperature will increase coastal ice-free areas, sea-ice loss, glacial retreat, ocean acidification, and ocean warming (Morley et al., 2020), affecting marine biota at all trophic level. Lower trophic levels are expected to move south depending on their tolerance to warming ocean condition and productivity; meanwhile, ocean acidification will mainly impact crustaceans and calcifying organisms. Marine mammals and seabirds are expected to move to alternative locations for food and breeding, survival, and adaptation (Constable et al., 2014). However, the impact of climate change on organisms and their associated microbiota as a whole (*i.e.*, the holobiont) has been poorly studied across the metazoan spectrum. Nevertheless, there are notable exceptions in some invertebrates and macroalgae (Table 1). Yet, we still lack ecosystem holistic approaches (including abiotic variables or ideally holobiont experimental assays where important abiotic parameters, such as temperature and pH, are controlled) that could give us insight into marine holobiont adaptability to the ongoing marine environmental changes.

Holobiont traits (*e.g.*, host functional traits and microbiome composition) might have a dynamic interaction with environmental changes. Host phenological changes, such as diet shifts, molting, and reproduction and seasonal changes might interact and cause seasonal microbial effects, which in turn, could impact nutrient recycling and biogeochemical cycles. For example, soils impacted by penguins and pinnipeds had high amounts of nutrients, such as carbon, nitrogen, and phosphorous (Ugolini, 1972; Tatur & Keck, 1990). Marine animal colonies impact in greenhouse gas emissions since penguin and pinniped settlements are hotspots for several greenhouse gases, such as $CO_2$, methane ($CH_4$), and nitrous oxide ($N_2O$) (Zhu et al., 2008, 2009). These animals hugely impacted the coastal sediments where they colonized through their feces, eggs, prey, carcasses, among others (Guo et al., 2018; Almela et al., 2022; Ramírez-Fernández et al., 2021). Feces directly impacted soil microbiome by seeding gut microbes from marine animals, and indirectly because of the high amount of nutrients that they transport from marine to terrestrial ecosystems (Guo et al., 2018). On a functional level, marine animals increased soil microbial communities related to denitrification pathways (Ramírez-Fernández et al., 2021) and other nitrogen pathways involved in $N_2O$ emissions.

Climate change studies should consider the impact of environmental changes in hosts and their microbiome. In the project on the surface microbiome of key species in the Strait of Magellan, we will measure environmental factors, such as water temperature, UV

radiation, chlorophyll content, oxygen levels, pH, salinity, and nutrient content, to correlate holobiont abundance, diversity, and distribution, focusing on the skin microbiome because it is in direct contact with the environmental parameters. Coupling environmental variables with holobiont data could aid in conservation efforts that protect both the host and its associated microbes and the ecosystem (*Carthey et al., 2020*).

## Surface microbiome of key species in the Strait of Magellan: an integrative holobiont project

To increase the number of microbial biodiversity studies in the Southern Hemisphere, we will conduct a microbiome project of key species in the Strait of Magellan in Chile (Fig. 4). The project will generate new baseline data for almost all the species that will be sampled (except for humpback whale skin, which has been extensively studied). Further the project will also generate microbiome functional data from multiple years (*i.e.*, metagenomics and metatranscriptomics) that will be integrated with environmental data. A longitudinal sampling approach will give valuable insights into host-microbiome responses to ongoing Anthropocene-derived climate change. Moreover, it will generate host microbiome data considering an ecosystemic approach, which will be valuable for further comparisons in later years when environmental variables would have likely changed.

Microbes are an excellent biodiversity study target since they can show fast evolutionary responses to environmental alterations and have enormous metabolic and genetic diversity. This project focuses on testing surface microbes as biosensors of climate change effects in key hosts in the Strait of Magellan. The project will encompass trophic-level inclusivity by sampling hosts in different (yet related) trophic levels and an introduced species (*i.e.*, farm salmon). The sampling design includes a primary producer (*i.e.*, *M. pyrifera*, huiro/kelp), primary consumers (*i.e.*, two crustacean species *L. santolla*, centolla and *M. gregaria*, channel prawns), secondary predators (*i.e.*, *S. fuegensis*, Fuegian sprat; *S. magellanicus*, Magellanic penguin; *A. patagonicus*, King penguin; and *M. novaeangliae*, humpback whale) and an apex predator (i.e*., O. byronia*, South American sea lion).

The project will generate original baseline knowledge of bacterial communities associated with the surface of some taxa (including penguin feathers, sea lion fur, fish scales, or crustacean shells). Furthermore, as multiannual samplings will be performed, the project will elucidate whether there are microbial signatures (both at the community and genomic levels) associated to seasonal variation and/or to environmental variables. One of the main hypotheses of this project is to test whether there is a core surface microbiome among marine hosts sharing the same environment (*i.e.*, specifically in the Coastal Marine Protected Area "Francisco Coloane"). Alternatively, other hypotheses have been considered, such as the existence of a core microbiome at different levels, for example, at host complexity (*i.e.*, a core microbiome for invertebrates, another for mammals, *etc.*); trophic level (*i.e.*, a core microbiome for primary consumers, another for primary predators, *etc.*); or alternatively a species-specific surface microbiome.

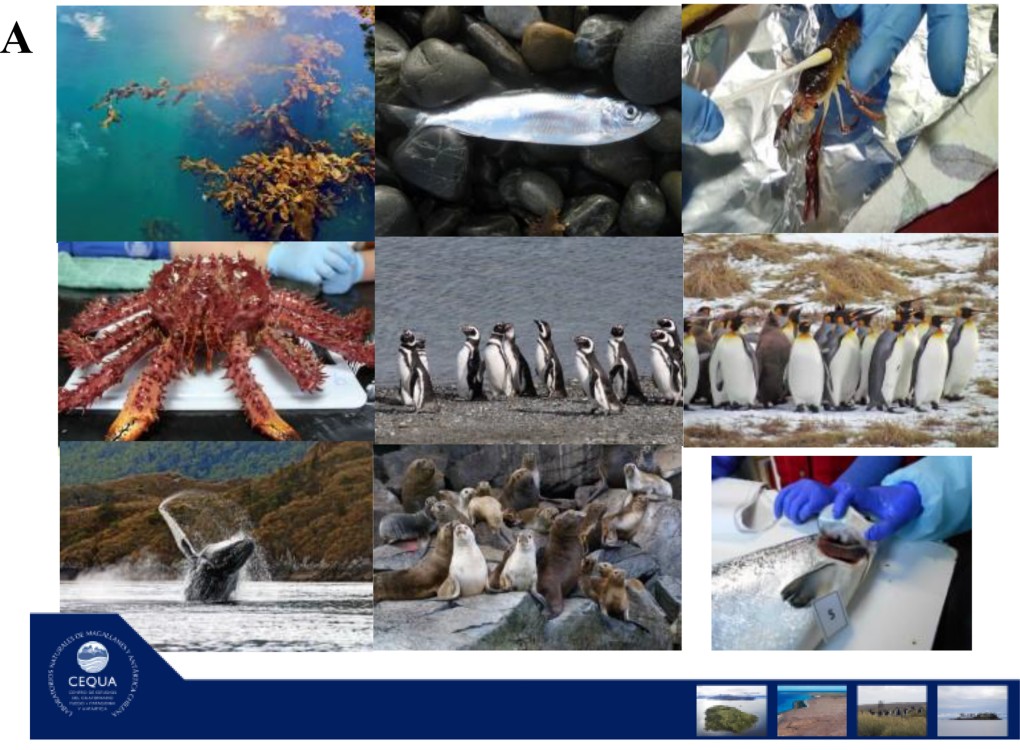

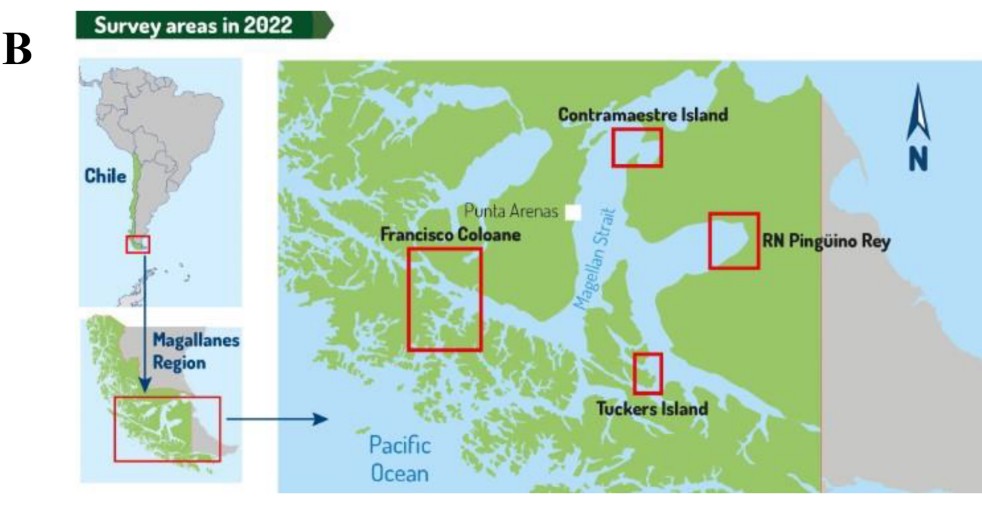

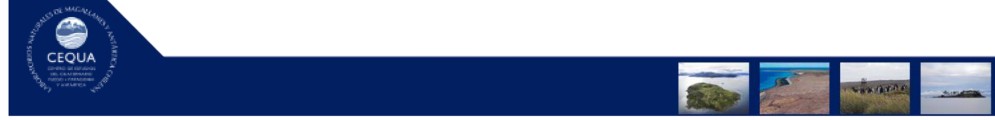

**Figure 4 Target species and main areas of field sampling of the Surface microbiome of key species in the Strait of Magellan.** (A) Target species of the microbiome project are (from top left to right): kelp/huiro (*Macrocystis pyrifera*), Fuegian sprat (*Sprattus fuegensis*), channel prawns (*Munida gregaria*), centolla (*Lithodes santolla*), Magellanic penguin (*Spheniscus magellanicus*), King penguin (*Aptenodytes*

**Figure 4** (continued)
*patagonicus*), Humpback whale (*Megaptera novaeangliae*), South american sea lion (*Otaria byronia*), Atlantic salmon (*Salmo salar*). (B) Main areas across the Strait of Magellan that field work is taking place. The lower right side (A and B) shows photographs of Carlos III Island, Contramaestre Island, King Penguin Reserve and Tuckers Islands.                

## DISCUSSION AND PERSPECTIVES

In general, the marine holobiont in the Southern Hemisphere has been represented by few systematic and authentic holobiont studies (*i.e.*, studies that analyzed host traits (*e.g.*, genomics, transcriptomics) and microbiome traits (*e.g.*, 16S, metagenome) at the same time, see Table 1 for examples of holobiont studies). Besides, there was a great bias in sampled hosts and research advance across geographic regions. For instance, sponges have been thoroughly sampled, but there was a great gap in marine invertebrate and vertebrate hosts. Moreover, we detect differences in microbiome research state of art methods across geographic regions. Functional microbiome and authentic holobiont approaches have been recently conducted in Australian macroalgae, cosmopolitan marine invertebrates, and sponges. Whereas in marine mammals, seabirds, and fish we still lack insights about functional microbiome and holobiont approaches. Interestingly, South America is the only region that has not performed functional microbiome approaches in any marine host.

Holobiont studies were dominated by microbiota approaches, where the focus was centered on taxonomic patterns across ecological/life story conditions. These studies have highlighted the prevalence of Proteobacteria among a wide spectrum of hosts, while specific host-phyla associations completed the bacterial community. Unfortunately, microbiota approaches have focused on the bacterial fraction, completely ignoring the potential role of fungi, virus, and archaea in marine host holobiont. Notable exceptions where these groups were studied are macroalgae (fungal communities), penguins (viromics), and sponges (archaea). Nevertheless, there were no studies addressing the interactions among all these groups simultaneously, likely because of technical and economic challenges (*i.e.*, computational resources, computational skills).

Microbiota studies are important and economically feasible explorations, yet detailed microbiome studies including metagenomic and metatranscriptomic studies are needed to inclusively address the microbe community (fungi, eukaryotes, virus, bacteria, and archaea) as well as its functional potential along with that of its host. Moreover, functional holobiont data (*i.e.*, host genomics and metagenomics/metatranscriptomics) coupled with environmental data might provide valuable insights about the influence of ecosystemic status in host-associated microbiome, which in turn could be tested for the plausibility of the microbiome to reflect host stress.

Microbiome studies might help to elucidate whether patchy/heterogenous distribution in microbe taxonomic profiles has different functional potential, or whether distinct taxonomic profiles have convergent/redundant functional profiles. Additionally, microbiome studies might help to elucidate whether there is a microbiome functional profile associated with eukaryote hosts, or whether there are core functions among their microbiomes even among distinct hosts. Finally, sampling must span as many individuals

as possible, as well as geographic and seasonal (longitudinal/annual) representability to determine whether patchy distributions among associated microbes are a natural feature of marine holobionts or a consequence of low sampling. Such a systems biology approach might bring further understanding of the complex interplay between microbes and marine hosts. We think that the project on the surface microbiome of key species in the Strait of Magellan will provide valuable information on the points mentioned above, which will contribute to the knowledge of microbial diversity in the region, as well as their current responses to Anthropocene-derived climate change.

## ACKNOWLEDGEMENTS

We are grateful to Rosalinda Tapia, Erika Aguirre, Claudio Moraga, Anelio Aguayo Lobo, Jorge Acevedo, and Lautaro Oyarzún for reviewing and helping to improve earlier versions of this manuscript. Additionally, we thank Gabriel Quilahuilque, Claudio Moraga, and Jorge Acevedo for providing Fig. 4. This article is part of the requirements for obtaining a Doctoral degree to Manuel Ochoa-Sánchez at the Posgrado en Ciencias Biológicas, UNAM.

### Funding

Financing was granted by CEQUA, project number RS0F0009 ANID, and a CONACYT Fellowship (CVU: 917392). This work was supported by ANID project number R20F0009. The funders had no role in study design, data collection and analysis, decision to publish, or preparation of the manuscript.

### Grant Disclosures

The following grant information was disclosed by the authors:
CEQUA: RS0F0009 ANID.
CONACYT Fellowship: CVU: 917392.
ANID: R20F0009.

### Competing Interests

Lía Ramírez-Fernández is employed by Centro de Desarrollo de Biotecnología Industrial y Bioproducto. E. Paola Acuña-Gómez & Valeria Souza are employed by Centro de Estudios del Cuaternario de Fuego. Manuel Ochoa-Sánchez is a doctoral student affiliated to Centro de Estudios del Cuaternario de Fuego. Luis E. Eguiarte & Valeria Souza are Academic Editors for PeerJ.

### Author Contributions

- Manuel Ochoa-Sánchez conceived and designed the experiments, performed the experiments, analyzed the data, prepared figures and/or tables, authored or reviewed drafts of the article, and approved the final draft.
- Eliana Paola Acuña Gomez conceived and designed the experiments, prepared figures and/or tables, authored or reviewed drafts of the article, and approved the final draft.

- Lia Ramírez-Fernández conceived and designed the experiments, performed the experiments, authored or reviewed drafts of the article, and approved the final draft.
- Luis E. Eguiarte conceived and designed the experiments, analyzed the data, authored or reviewed drafts of the article, and approved the final draft.
- Valeria Souza conceived and designed the experiments, analyzed the data, authored or reviewed drafts of the article, and approved the final draft.

## Data Availability

The database for maps is available in the Supplemental File.

## Supplemental Information

Supplemental information for this article can be found online at http://dx.doi.org/10.7717/peerj.15978#supplemental-information.

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
