# Peer review of "Current knowledge of the Southern Hemisphere marine microbiome in eukaryotic hosts and the Strait of Magellan surface microbiome project"

_PeerJ, doi:10.7717/peerj.15978_

## Round 0.1 · original submission · Major Revisions

Going through the reviews' comments and also my personal opinion, the revised manuscript should be substantially reorganised. As it is now the focus on the southern hemisphere and the Magellan Straits seem to have been chosen with no clear criteria. Instead, a focus on one or only a few hosts in both hemispheres would give more value and interest to this paper. Such an approach it would show, why the organisms' microbiomes behave differently in the Southern vs. Northern hemisphere, especially when many of the organisms discussed in the current version are cosmopolitan.

Reviewer 1 ·

Basic reporting

The manuscript is very well written, clearly, with adequate language and adequate English translation.

The references are enough and support the information compiled by the authors on the studies on the microbiota of the Southern Hemisphere.

Sufficient introduction and context are provided, they are presenting the characteristics of this particular area and highlight the gaps in the related information. Something very interesting is also the approach of the project in the Strait of Magellan, an area little explored and with a lot of potentials to generate new knowledge.

Higher quality is needed in the figures, I see the maps as stretched, Figure 1 does not include the legends of the graph axes. The legends of Figures 3 and 4 cannot be read clearly.

Experimental design

The authors do an extensive search and gather a good amount of research that enriches this review. The sources are adequately cited. The review is organized logically into coherent subsections.

Validity of the findings

The main argument of this review is well developed and supported with references, therefore it meets the objectives established in the introduction, making a comprehensive review of the existing knowledge about marine holobionts and their microbiota in the Southern Hemisphere, however, I would have liked to see the project on the microbial communities of the Strait of Magellan further developed.

In conclusion, considerations on unresolved issues, gaps in research in this particular area, and the future directions addressed in their project on the Strait of Magellan are included.

Line 371 requires a space between is16S

Additional comments

I consider that the manuscript can be published as it is written, however, it would be recommendable to improve the figures.

Reviewer 2 ·

Basic reporting

The review aims to compile microbiome research from the Southern Hemisphere into a single review compilation. The review discusses all identified studies on macroalgal, invertebrate, seabird, and mammal microbiomes. Another aim of the review as defined by the authors was to discuss a new study proposed to be conducted by the researchers.

The aim of synthesizing the status of the field and gaps in knowledge for microbiome studies in general is a valuable one. Incorporating and discussing all marine microbiomes in the entire Southern Hemisphere appears a large, perhaps impossible task. This approach comes with the caveat that the treatment by each host group is somewhat superficial. Another angle that could have been taken would have been to focus on one host organism group and examine distributions and knowledge in Southern hemisphere habitats and species in the context of global distributions and understanding of mechanistic controls. I expect the review could reach more depth that way. In some cases, some references are made to studies from Northern Hemisphere, but given the large scope, it cannot be done comprehensively for all included host groups. As such, a more broad overview to Southern Hemisphere microbiomes in general seems fine, but given the large scope (all hosts, entire Southern Hemisphere), the necessarily broad overview-level discussion leaves this reader looking for attempts to elucidate mechanisms more in-depth. In my opinion, an in-depth review of a single host group globally would be more valuable and interesting than a more superficial view into highly diverse organisms in one hemisphere only. Why do we expect the organisms microbiomes behave differently in Southern vs. Northern hemisphere?

The part about proposed project not yet conducted should be completely cut. This part makes the review read like a grant proposal - perhaps it was meant to be one or was one previously. A review can and should certainly highlight gaps in knowledge and synthesize understanding. Here, proposing a specific study to address these gaps in knowledge seems out of place. Some of the content in the discussion section where the authors discuss the purpose and aims of the proposed new study could still be incorporated in a form that reads more broadly as aims that remain open for future work.

The review heavily focuses on the holobiont concept, the terminology of which has been challenged (e.g. Douglas and Werren 2016. Holes in the hologenome: why host-microbe symbioses are not holobionts. mBio). As the authors here also elude to, presence of microbiome members often cannot be directly linked to host biology, but may assemble from the environment randomly. In addition, competitive and facilitative interactions may persist in the microbiomes. Not all microbiome members fall under the 'holobiont' context from the perspective of known influence on organismal function. I would strongly recommend the authors discard the referrals to somewhat dated view of 'holobiont' and simply refer to microbiomes.

Experimental design

The approach was to conduct a literature review with a comprehensive inclusion of all studies in Southern Hemisphere. The authors used a list of keywords that appears to have targeted different types of organisms known to have microbiome data available, but it leaves some questions whether this was sufficient. It may be the keyword list limited some findings (e.g. fishes, given only 'marine vertebrate' was included as a keyword). Other keywords might have provided additional hits, such as 'macroalga*', 'coral', 'cnidarian', 'copepod', 'krill', 'jellyfish'. The authors should acknowledge that the selected list of keywords may have limited the scope of the findings. Figure 1 is unnecessary. The numbers can be reported in one sentence in the text.

Validity of the findings

In my view, sections of the discussion should be significantly tuned down and cut back. Here the authors are making an attempt to explain the microbiome variability throughout the diverse habitats and hosts in the entire Southern Hemisphere.
L1001 "We think..." Should be reworded. A review should take a view of synthesizing and coming up with insights based on data provided. The section L1001-L1015 is not well integrated and comes off a bit speculative. L1025-1063 should be deleted (along with sections in other paragraphs directly referring to the proposed new study). The discussion should be revised to read more broadly rather than as a justification for a specific new study proposed. Other current sections in the discussion could serve a more broadly written revised conclusions section.

Additional comments

The narrative generally reads ok, but there are issues in English, some of which are occurring repeatedly.

plural-singular issues:
sponge microbiome - not sponges' or sponges microbiome. Applies also for pinniped (not pinnipeds') microbiome, coral (not corals') microbiome, bird (not birds') microbiome etc. Also, microbiota, not microbiotas
... is a gammaprotebacterium (not ...bacteria)
L 438, 450 sponge-assiciated (not sponges associated)

Generally, place the subject before verb. For example, L84-85 should read: "As climate change progresses, an intensification in seasonality is expected, which might intensify..."

There are quite a few issues in punctuation, which makes the text difficult to read at times. Don't separate clauses with both comma and 'and'. Make sure each clause has a verb. Use adverbs properly.
Examples of incorrect use of adverbs:
L89 These include,...
L265 Although,
L275; 326 Besides,
Other examples of incorrect comma (these are examples, not a comprehensive list):
L78-79 ..warmer temperatures coupled with low to moderate winds, increase ice melt, which...
Other points with comma issue(s) include: L 268, 286, 327, 334, 336, 381, 494, 530, 779, 781, 783, 784, 868, 909, 972

L309 Presence of a low diversity microbiome does not necessarily equate to 'gardening'.
L311-313 But comparisons were not made to macroalgae in other parts of the world, so the conclusions remain incomplete.

The narrative breaks down at times. Start paragraphs with a topic sentence that ties the paragraph to the entire narrative. Example of a topic sentence that should be improved:
L78: "However, there are also some surprising results".

Other points
L65 plankton, not plancton
L70 rate is, not 'are'
L73 chlorophyll a (a italicized)
L217 '16S' - when referring to it first time write out '16S rRNA gene' and define that thereafter '16S' is used. The 'S' should always be capitalized (see Table 1).
L226 delete 'species of'
L240 fix position of 'conducted' in sentence
L245 ', being the' - correct grammar in sentence
L270 How are diatoms pathogens? Are you referring to domoic acid producers?
L408 Cellulophaga algicola is a bacterium, not a sponge
L520 sulfate reducer?
L589 is > are
L612 template? do you mean in temperate?
L755 what about potential acidity in stomach?
L790 occur
L794 with 'their' are you referring to viruses? The potential of marine animals for serving as a reservoir for potential human pathogenic viruses is relevant in this context (with published evidence existing for at least marine mammals)
L859 in>on

Figure 2. Template? Do you mean Temperate?
Figures 2-3. The yellow dots are difficult to discern from a printed page. Light blue also tends to disappear to the green land masses. Use more distinct colors.

---

## Round 0.2 · Minor Revisions

There are a few more things to by answered. Please provide again a point-by-point rebuttal to all of the comments.

Reviewer 2 ·

Basic reporting

The authors have made some edits to the manuscript. The figures are improved.

The abstract should be improved for readability. For example, the abstract mixes various tenses, sometimes within a sentence, which makes it difficult to follow.

The focus is intended to be on Southern Hemisphere, but at times text jumps to the Northern hemisphere. The text and narrative flow could be tightened and focused throughout for better readability.

Issues on punctuation remain in the manuscript (only some of them have been indicated below). I made the point about the punctuation issue in my previous review and provided examples. In addition, I have indicated a few points where the meaning was lost due to a grammar issue.

Experimental design

The authors chose a broad scope and used somewhat limited keywords.

Validity of the findings

The authors provide some information on summarizing studies conducted on microbiomes of organisms in the southern hemisphere. This should provide some value to the research community.

Additional comments

Line numbers below refer to the lines in the tracked changes Word document.

L40 add comma after ‘Yet’
L46 beside >besides
L46-47 tense changes within in one sentence
L50-51 microalgae is plural
L57 ‘the project’ here is not clear, as the reader at this point is not familiar with the project. ‘a future project’? ‘a proposed project’? ‘an ongoing project’? I don’t know what you are referring to here.
L222 chlorophyll a (with a in italics). Remove the dash (chlorophyll-a).
L300-301 macro-organismal diversity
L383 Delete comma from: “that microbiomes, and holobionts”
L384-386 sentence is missing a word or words
L434 revise ‘assemble’ to ‘assembly’
L437 revise ‘surface’ to ‘epibiotic’
L442 revise ‘host’s’ to ‘host’
L516 revise ‘on the one hand’ to ‘on one hand’
L522-524 I suggest revising the sentence for grammar
L529-530 Please clarify the nature of the ‘long term project’. Is it ongoing? Planned? Proposed?
L531 Revise ‘Later, we’ to ‘We then…’
L531-533 please rewrite to correct grammar
L541 Please clarify if ‘Several keywords were used…’ means that the following keywords you list are explicitly all keywords used. I suggest replacing ‘several’ with ‘following’.
L607 Remove comma after ‘corals’
L612-617 very long sentence with punctuation issues. Please break into two to improve readability.
L620 When referring to many fish species, the term is ‘fishes’. Please correct throughout.
L628-629 suggested revision to: ‘we did not identify any studies in which South American marine hosts were studied with metagenomics’
You need to acknowledge here the fact it is possible that not all studies were found with the keyword selection that was used. Be careful declaring South America void of metagenomics studies on microbiomes. It is possible you missed some.
L635-636 From the fact that there are many macroalgal species it does not necessarily follow that there is a large diversity that remains unexplored. Unless you also first state that much of the diversity is known to be unstudied, ‘Therefore’ is incorrect. Please reword for accurate statement.
L690 unclear what you mean by ‘DNA holobiont’
L690, 1302 ‘worldwide’ is confusing here, since previously you stated you are restricted to the Southern hemisphere. If you say ‘worldwide, predominantly in Australia, Antarctica, and Southern America’, it sounds like these are highest density areas also considering the northern hemisphere. The meaning of the sentences is unclear. Are we considering sites worldwide with this qualitative assessment? This is confusing.
L698 ‘nevertheless’ does not fit here. The previous sentence does not negate the fact studies were also done in the Southern Hemisphere.
L712 1. It is unclear what you mean by this: “Nevertheless, at lower taxonomic levels (genus/Amplicon Sequence Variant [ASV]/ Operational Taxonomic Unit [OTU]), there is a rich spectrum of valuable microbes (Egan et al., 2013; Hollants et al., 2013).”
2. In addition, what do you mean by ‘valuable’?
L967 host trait measurement should not be equated to transcriptomics – this is much broader concept
L968 This is unclear/grammar needs revision: “could aid further to elucidate if the macroalgae is actively filtering its microbiome and fitness relevance.”
L1491 Punctuation and grammar issues make this sentence very difficult to follow: “Noteworthy, phylogenetic novelty varied across sponge-associated bacteria genera; whereas Sporosarcina and Nesterenkonia had the greatest phylogenetic novelty, other sponge-associated bacteria, such as Cellulophaga algicola, had more known strains (Moreno-Pino et al., 2021).”
L1720 acyl-homoserine-lactone
L1806 Correct grammar in the format “Antarctica sponge” used here and elsewhere in the narrative.
L1869 Once again, it is confusing to the reader if you are referring to the entire marine environment or only the Southern hemisphere.
L2181, 4124 revise ‘Noteworthy,’ (grammatically incorrect)
L2188 bacterial
L3648 Weddell
L3670 You need to add a comma after ‘Yet’ (here and elsewhere in the text)
L3754 the microbiome project > a microbiome project

Figure 1
The figure still shows ‘Template’

Figure 4A
The information of the species could be provided in the text. The value of the figure is primarily decorative.

---

## Round 0.3 · accepted · Accept

The paper can now be accepted after all the careful revisions have been taken into account.